# Slowest possible replicative life at frigid temperatures for yeast

Diederik S. Laman Trip[1,2], Théo Maire[1,2] & Hyun Youk ●[2,3] ✉

Determining whether life can progress arbitrarily slowly may reveal fundamental barriers to staying out of thermal equilibrium for living systems. By monitoring budding yeast's slowed-down life at frigid temperatures and with modeling, we establish that Reactive Oxygen Species (ROS) and a global gene-expression speed quantitatively determine yeast's pace of life and impose temperature-dependent speed limits - shortest and longest possible cell-doubling times. Increasing cells' ROS concentration increases their doubling time by elongating the cell-growth (G1-phase) duration that precedes the cell-replication (S-G2-M) phase. Gene-expression speed constrains cells' ROS-reducing rate and sets the shortest possible doubling-time. To replicate, cells require below-threshold concentrations of ROS. Thus, cells with sufficiently abundant ROS remain in G1, become unsustainably large and, consequently, burst. Therefore, at a given temperature, yeast's replicative life cannot progress arbitrarily slowly and cells with the lowest ROS-levels replicate most rapidly. Fundamental barriers may constrain the thermal slowing of other organisms' lives.

An important question is how countless biochemical reactions collectively dictate life's pace. There is the familiar, loosely defined notion of life progressing at some rate towards death. But a major challenge is rigorously defining and quantifying this rate, even for one cell, and then determining how each intracellular process affects this rate. From the perspective of physics, resolving this conceptual challenge would advance our understanding of how living cells keep themselves out of thermal equilibrium. Of particular interest is determining whether constraints exist on the rate at which a cell's life can progress. Such constraints would mean that there are physical limitations to the cell's ability to keep itself out of thermal equilibrium by using nutrients to maintain and build a copy of itself (i.e., divide into two cells). Without invoking any molecular mechanisms, one can reason that a cell cannot take an arbitrarily short duration to self-replicate. But it is unclear, even at the level of a conceptual reasoning that does not require molecular details, whether a cell cannot take an arbitrarily long time to self-replicate. For example, would it be possible for a cell to take 10 years to progress through the cell cycle without stopping (i.e., have a doubling time of 10 years)? How would intracellular processes drive life at such a slow pace? If not, then why not? By using the budding yeast (*S. cerevisiae*) as a model system, we sought to quantify the pace at which each cell can traverse through the eukaryotic cell cycle and then determine whether this pace could be arbitrarily slow. We studied yeast at frigid temperatures (0–5 °C) at which intracellular processes are expected to be extremely slow, though the exact speed remains unknown for many processes[1]. Frigid temperatures are also relevant for understanding organisms that cannot regulate their internal temperatures—such as microbes, plants, and cold-blooded animals—that frequently inhabit frigid environments. Researchers have discovered specific genes, stress responses, and epigenetic mechanisms that contribute to sustaining a cell's life in frigid environments[2–4]. But, beyond a few specific mechanisms[1,5–10], we currently do not know how an interplay of myriad processes dictate and potentially constrain a cell's progression in life at frigid temperatures.

It is unclear whether and how the budding yeast can live at low temperatures due to conflicting observations. Two popular views

[1]Kavli Institute of Nanoscience, Lorentzweg 1, 2628CJ Delft, The Netherlands. [2]Department of Systems Biology, University of Massachusetts Chan Medical School, 368 Plantation Street, Worcester, MA 01605, USA. [3]CIFAR Azrieli Global Scholars Program, CIFAR, 661 University Ave. Suite 505, Toronto, ON M5G 1M1, Canada. ✉e-mail: hyun.youk@umassmed.edu

argue that yeast cannot survive and/or proliferate at sufficiently low temperatures. One view proposes that essential processes such as transcription and molecular transport are too slow for sustaining cell proliferation at sufficiently low temperatures[1,8]. A complementary view proposes that yeast cannot proliferate at sufficiently low temperatures due to physical damages caused by their cell membranes becoming too rigid, proteins denaturing, oxidative stresses, and other events[1,5,9,10]. Yeast can repair such damages by, for example, expressing genes to fluidify their membranes, (re-)folding proteins with chaperones, and responding to oxidative stresses[1,8–10]. But at sufficiently low temperatures, the damage from various sources is thought to be too severe that yeast cannot survive and divide[5,9,11]. Importantly, how much each type of damage is responsible for causing yeast to die at frigid temperatures is unclear (i.e., exactly why a yeast fails to divide and survive at sufficiently low temperatures, and what determines that a temperature is sufficiently low, remains unknown). Despite the two popular views mentioned above, daily experience shows that some baker's yeast cells in an isogenic colony can evidently proliferate, albeit slowly, when stored in a refrigerator, and that fungal colonies can appear on refrigerated foods after months. Moreover, cellular processes can remain active despite being slow[6,8,11]. These daily-life observations, however, are often made in settings in which temperature is not well controlled. They do not reveal rigorous insights or any constraints on a cell's life in frigid environments.

Our study starts with a discovery of yeast cells helping each other live at frigid temperatures: they secrete and extracellularly accumulate glutathione—an antioxidant—to inactivate harmful Reactive Oxygen Species (ROS) which naturally arise and are the key cause of death for yeast at frigid temperatures. We found how the abundance of intracellular ROS determines yeast's ability to grow, divide, and survive at frigid temperatures by continuously monitoring individual cells for weeks to months and by using single-cell-level analyses. We found that all these effects of ROS are due to one mechanism: ROS elongates the G1 (growth) phase of the eukaryotic cell cycle, thereby continuously increasing the cell size while inhibiting the entrance into the S-G2-M (replicative) phase. Crucially, we find that frigid temperatures impose a threshold concentration of ROS, with a corresponding threshold amount of time for staying in G1, so that cells with more than an above-threshold amount of ROS (staying longer in G1 than the threshold duration) burst without dividing. An interplay of ROS and global gene-expression dynamics determines the threshold duration for exiting G1 at frigid temperatures. A simple mathematical equation, exhibiting a power–law relationship, summarizes our findings into a coherent picture: a cell can self-replicate by completing the cell-cycle at any speed that lies between two speed limits—shortest and longest allowed doubling times—that are specific to each temperature. Crucially, our work suggests that the longest allowed doubling time increases without an upper bound as the temperature decreases towards the freezing point of the growth medium. Hence yeast's life can be slowed down to an arbitrarily slow pace. However, we found that such ultra-slow self-replication, while possible, becomes exceedingly unlikely as temperature approaches the freezing point. Together, these results reveal quantitative limits to slowing-down self-replication dynamics at frigid temperatures.

## Results

### Cells help each other survive at frigid temperatures
At frigid temperatures, we observed liquid cultures of a laboratory-standard yeast strain at various population densities, ranging from 10 cells per ml to $10^5$ cells per ml. We incubated these cultures of wild-type yeast at various fixed temperatures (4 °C to 14 °C) in a high-precision, thermostatic incubator that maintained a desired temperature within ±0.1 °C (Supplementary Fig. 1). During 2 months of incubation, we regularly took aliquots from each culture to measure its population

density with a flow cytometer. At temperatures above 6 °C, population of every density slowly grew (Supplementary Figs. 2, 3). But at 6 °C, surprisingly, none of the populations that started with 10 cells per ml grew whereas populations that started with at least 25 cells per ml grew to reach the carrying capacity during the 2 months. Similarly, at 5 °C, populations that started with at least 3000 cells per ml grew (Fig. 1a) whereas none of the populations that started with 1000 cells per ml or less grew at all during the 2 months (Fig. 1a). Strikingly, at 4.7 °C—just 0.3 °C below 5 °C—no populations grew regardless of their initial density. In fact, at any temperature below 4.7 °C, we did not observe any populations growing regardless of their initial density. Combining these results yielded a phase diagram that indicated for which temperature and initial population-densities a population could grow (Fig. 1b). This diagram showed that the density-dependent population growth only occurs within a narrow, 1 °C-window between 5 and 6 °C. It also indicated that 5 °C is the lowest temperature at which yeast populations can grow (Supplementary Figs. 4, 5). But, as we will show, the phase diagram hides the yeast's true ability to duplicate at even lower temperatures (e.g., 1 °C).

To elucidate the origin of the density-dependent population growth at 5 °C, we used a wide-field microscope to continuously monitor individual cells in high-density (6250 cells per ml) and low-density (250 cells per ml) populations for 3 weeks at 5 °C. The high-density population grew towards the carrying capacity whereas the low-density population did not grow at 5 °C. For each cell, we determined whether it eventually died, stayed alive and duplicated, or stayed alive without duplicating (i.e., continuously increased in size) during the 3 weeks (Fig. 1c). We found that most cells in the high-density population survived whereas most cells in the low-density population died. Hence, the low-density population was headed towards extinction. Moreover, while both populations had only a minority of cells that divided, the low-density population had fewer dividing cells (17% duplicated) than the high-density population (29% duplicated). The duplicating cells had doubling times that ranged from 2 days to 17 days in both populations (Fig. 1c). Altogether, these results establish that the density-dependent population growth at 5 °C arises from cells helping each other to survive and duplicate (Supplementary Notes).

### Cells secrete glutathione to remove reactive oxygen species
We next asked how yeast help each other in surviving and dividing at low temperatures. At 5 °C, a supernatant of the high-density population induced growth of the low-density population that would have gone extinct without the supernatant (Supplementary Fig. 6). Hence, yeast cells were likely secreting molecules that promoted their duplications. Motivated by our previous work[12] which showed that yeast start to secrete and extracellularly accumulate glutathione, an antioxidant, as temperature increases above 37.5 °C, we hypothesized that yeast also secretes glutathione at sufficiently low temperatures. Indeed, at 5 °C, we detected glutathione gradually accumulating in the growth medium of high-density (growing) populations but not for low-density (non-growing) populations (Fig. 2a and Supplementary Fig. 7). In fact, at every temperature below 8 °C, we found that populations accumulated glutathione as they grew (Supplementary Fig. 8).

Glutathione is yeast's key antioxidant[13,14] that is involved in inactivating (reducing) ROS. ROS can damage cellular components (e.g., nucleic acids, proteins, and cell membranes[15–17]). Hence, we used live-cell fluorescent reporter dyes to measure the ROS concentration inside each cell of the high-density (growing) population and the low-density (non-growing) population. We found that cells of the high-density population typically had much less intracellular ROS than cells of the low-density population (e.g., ~10-fold less superoxides) (Fig. 2b). These results establish that lower intracellular ROS concentrations are associated with population growth.

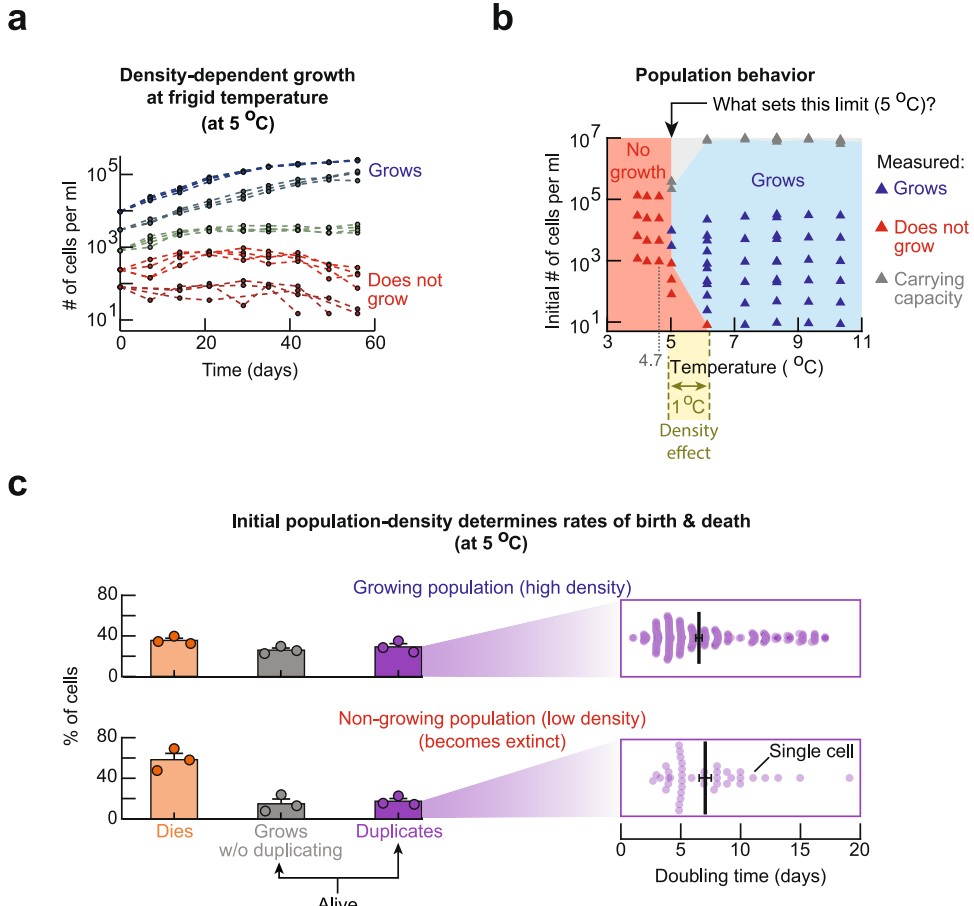

**Fig. 1 | Yeast cells help each other in surviving and duplicating at frigid temperatures. a** Population density (number of cells per ml) of wild-type yeast over time, measured with a flow cytometer at 5 °C. Each curve shows a different population (*n* = 4 biological replicate populations). Blue: populations that grow to the carrying capacity. Red: populations that do not grow. **b** Phase diagram that summarizes all growth experiments of type shown in **a**. Each triangle represents at least *n* = 4 biological replicate populations that exhibit the same behavior as shown in **a**. **c** Result from monitoring individual cells of two populations for 20 days with a wide-field microscope. Upper half is for a growing, high-density population (initially ~6250 cells per ml). Lower half is for a non-growing, low-density population (initially ~250 cells per ml). Bars represent mean of *n* = 3 biological replicates with s.e.m. Zoomed out boxes on the right show doubling times of individual, duplicating cells (each purple dot represents one cell). Black bars denote average doubling time: 6.5 ± 0.3 days for the high-density population and 7.1 ± 0.5 days for the low-density population (error bars are s.e.m., for *n* = 3 biological replicate populations). The doubling time for one cell is the time elapsed, from the moment when a bud appears on a mother cell to the moment when either the daughter (after being born) forms its own bud or when the mother forms another bud. Data also includes cells whose divisions were unfinished at the end of the time-lapse movie. All data in this figure are in the Source Data file.

## Reduced glutathione enables and speeds up cell duplication

Since glutathione accumulates only for a growing population and lower levels of intracellular ROS are associated with growing populations, we hypothesized that ROS is the primary inhibitor of cell division for yeast at frigid temperatures instead of being a mere correlative measure. To test this idea, we added a high concentration (50 μM) of reduced glutathione (GSH) to the growth media of a low-density (non-growing) population at 5 °C. The added GSH caused the low-density population to grow (Fig. 2c) whereas the same, low-density population did not grow without the added GSH (Fig. 2c). Moreover, the added GSH decreased the ROS concentrations in cells of the low-density population (Fig. 2d) to nearly the same levels seen in cells of the high-density (growing) population (Fig. 2b and Supplementary Fig. 9). These results establish that GSH alone can decrease ROS concentrations in cells of low-density (non-growing) populations and induce cell proliferation at 5 °C. Specifically, we determined that the low-density populations must be given at least ~1 μM of extracellular GSH to grow at 5 °C (Supplementary Fig. 10). Moreover, we found that population growths arise from GSH's role as an antioxidant (Supplementary Fig. 11) and not from GSH's other, known roles.

We further discovered that increasing the extracellular GSH concentration accelerates population growth at 5 °C (Fig. 2e). For

example, without adding any GSH, the low-density populations did not grow whereas sufficiently high-density populations doubled in density once every 7–10 days while accumulating up to ~0.1 μM of extracellular GSH (Fig. 2e). Low-density populations, whose growth media were supplemented by GSH, could grow with a doubling time ranging from 6 days (with 1 μM of GSH) to 3 days (with 1 mM of GSH) (Fig. 2e). Thus, we could accelerate and tune the speed of population growth by varying the amount of extracellular GSH. We further confirmed this by monitoring, with a microscope, individual cells over weeks. We found that giving ample GSH to the low-density population more than tripled the percentage of cells that duplicated to 70% and shortened the average doubling time of a cell by more than half (Supplementary Fig. 12).

Together, the above results revealed the cooperative mechanism by which yeast cells survive, duplicate, and avoid population extinctions at frigid temperatures: yeast cells collectively build an extracellular pool of GSH that they then use to reduce each other's intracellular ROS. Importantly, above results establish that ROS is the primary inhibitor of cell duplication for yeast at frigid temperatures.

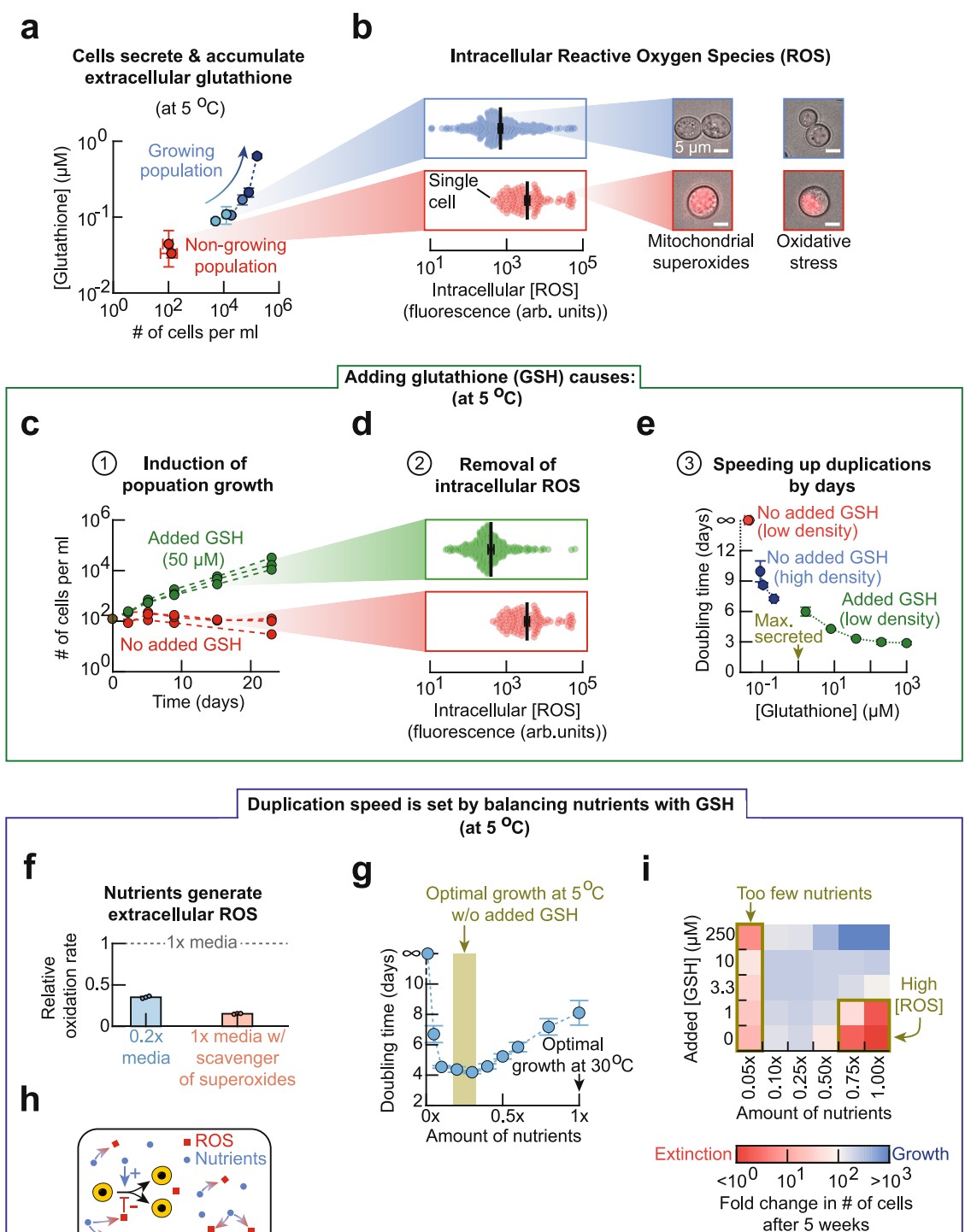

**Fig. 2 | Reduced glutathione (GSH) removes reactive oxygen species (ROS) to enable cells to duplicate at frigid temperatures. a** At 5 °C, concentration of extracellular glutathione for non-growing, low-density populations (red: initially ~300 cells per ml) and growing, high-density populations (blue: initially >900 cells per ml). Arrow indicates time passing during population growth. Data shown as mean ± s.e.m.; $n = 3$ biological replicates. **b** Intracellular ROS proxied by a fluorescent dye for mitochondrial superoxides, measured after 2 weeks of incubation at 5 °C. Each dot is one cell from a high-density population (blue: ~6250 cells per ml initially) or a low-density population (red: ~250 cells per ml initially). Black data represent mean ± s.e.m.; $n = 3$ biological replicate populations. Representative pictures shown. Left column: fluorescent dye for mitochondrial superoxides. Right column: fluorescent dye for general cellular ROS. Scale bar: 5 μm. **c** Populations incubated with (green) or without (red) 250 μM of reduced glutathione (GSH) (all initially ~120 cells per ml). **d** Intracellular ROS level in cells after 2 weeks of incubation, corresponding to populations shown in **c**. Each green and red point represents one cell. Black data represents mean ± s.e.m.; $n = 3$ biological replicate populations. **e** Red: low-density population (initially ~250 cells per ml) without supplemented GSH. Blue: high-density populations without extra GSH supplemented. Green: low-density population (initially ~250 cells per ml) that grows after receiving GSH. Data shown as mean ± s.e.m.; $n = 4$ biological replicates. **f** Oxidation rate in a 0.2× medium (blue) or in 1×-medium with 100 μM of Trolox (scavenger of superoxides) (purple). Rate normalized to oxidation rate of 1x-medium. Data shown as mean ± s.e.m.; $n = 3$ biological replicates. **g** Population's doubling time (initially ~190 cells per ml). Data shown as mean ± s.e.m.; $n = 3$ biological replicates. **h** Summarizing (**f**, **g**). **i** Each pixel represents average fold-change in population density after 5 weeks of incubation with indicated amount of GSH and nutrients (all started with ~210 cells per ml). $n = 3$ biological replicates. All data in this figure are in Source Data file.

## ROS-generating nutrients affect cell-duplication speed

We next sought to identify a major source of ROS which we hypothesized to be some of the extracellular nutrients. We used the same fluorescent, oxidation-responsive dye as before to measure ROS levels in the growth medium (1× medium). The growth medium consisted of ample (2%) glucose and non-sugar nutrients that are common to yeast and other microbes (i.e., essential amino acids, vitamins, etc.). How rapidly the fluorescence of the dye changed over time (i.e., oxidation rate) in media without any yeast was a proxy for the ROS-creation rate by the media components, as demonstrated by the fact that adding scavengers of ROS (superoxides) to the 1× medium caused a sixfold decrease in the oxidation rate at 5 °C (Fig. 2f). We found that the non-sugar nutrients, but not glucose, were the primary generators of ROS in the media at 5 °C (Supplementary Fig. 13). Decreasing the amount of non-sugar nutrients by 80% caused a ~85% decrease in the oxidation rate in the medium (Fig. 2f and Supplementary Fig. 13) and shortened the population's doubling time from 8 days to 4 days at 5 °C (Fig. 2g). Decreasing the non-sugar level further, however, sharply increased the population doubling time (Fig. 2g). Hence the non-sugar nutrients have dual, opposing roles: they both promote and inhibit cell proliferation (Fig. 2h). Incubating yeast with varying amounts of GSH and non-sugar nutrient revealed the full extent to which we could inhibit or enable population growths at 5 °C (Fig. 2i and Supplementary Fig. 14). Together, these results show that non-sugar nutrients are major generators of ROS in frigid temperatures. We also found that limiting aeration—lowering $O_2$ abundance—prevents populations from growing at 5 °C (Supplementary Fig. 15). Hence, like the ROS-generating nutrients, $O_2$ is required for yeast populations to grow at 5 °C, thereby highlighting the inevitability of generating ROS at 5 °C.

## ROS inhibits cell division and promotes death

With ROS being the primary inhibitor of population growth, we sought to elucidate how ROS affects an individual cell's ability to divide, grow, and survive. To do so, we used a wide-field microscope to continuously monitor individual cells for 3 weeks at 5 °C. At the start of the 3-week period, we used a fluorescent ROS-reporter dye as in Fig. 2b to determine the ROS concentration in each cell. Subsequently, over the next 3 weeks, we determined which of the following four events pertained to each cell: cell began as a bud at the start of our observation period and successfully separated from its mother to begin its life (labeled "begins life" in Fig. 3a); cell duplicated (labeled "duplicates" in Fig. 3a); cell's size continuously increased without a bud ever appearing (labeled "grows" in Fig. 3a); and cell died (labeled "dies" in Fig. 3a) (Supplementary Movies 1, 2). We found that every cell continued to increase in its size over time, regardless of which of the above events was occurring. Moreover, we observed that virtually all cells that died did so by bursting. By determining what percentage of cells underwent each of the four events for a given concentration of intracellular ROS, we found the probability of each event occurring as a function of intracellular ROS concentration. We first did so for cells of a population named, "low [GSH] population". This was a high-density population that grew at 5 °C by accumulating relatively low amounts of GSH (0.1–1 μM) (Fig. 2a). In this population, cells with more ROS were less likely to divide and more likely to die (Fig. 3b). Hence ROS decreases the chance of dividing and increases the chance of dying.

Similarly, we determined the probability of each event occurring as a function of ROS level for a population named, "high [GSH] population" (Fig. 3b). This was a low-density population to which we gave an abundant (250 μM) GSH to induce its growth at 5 °C. Despite the high GSH concentration, we found that cells of the high [GSH] population spanned the same range of ROS levels as the cells of the low [GSH] population (i.e., 10–$10^5$ fluorescence units in Fig. 3b). This suggested that, even with the high amount of GSH that we gave to this population, cells cannot reduce ROS when ROS is sufficiently

abundant. Indeed, cells with more than some threshold concentration of ROS in the high [GSH] population had a virtually zero chance of dividing (i.e., purple curve ends at ~6000 arb. units of ROS in the bottom graph in Fig. 3b). Cells with an above-threshold concentration of ROS could still increase in size or, more likely, die. As in the low [GSH] population, we found that cells with more ROS were more likely to die and less likely to divide. However, while having a less-than threshold concentration of ROS did not guarantee that the cell would divide in the low [GSH] population, it virtually guaranteed that the cell would divide in the high [GSH] population (Fig. 3b).

## Larger cell is more likely to die and less likely to divide

We further established that the amount of ROS in a cell does not correlate with the cell size (maximum cross-sectional area) (Supplementary Fig. 16). Hence, we reasoned that cell size, like ROS, may serve as a predictor of a cell's future[18–22]. We tested this idea by continuously monitoring how each cell's size changed over 3 weeks at 5 °C with a microscope. We could then determine how likely a cell of a certain size engaged in each of the four events mentioned above (Fig. 3c). We examined both the low [GSH] and high [GSH] populations. In both populations, we found that larger cells were more likely to die and less likely to divide (i.e., less likely to form a bud to start cell division) (Fig. 3c). In both populations, cells of all sizes had nearly the same chance of increasing further in size without dividing (Fig. 3c). By comparing the two populations, we discovered that increasing the GSH level—and thereby decreasing the average ROS level in a cell—greatly increased the chance of dividing for cells of nearly all sizes (Fig. 3c). In fact, reducing the average ROS level caused even some of the largest cells to divide.

## ROS cuts replicative and elongates chronological lifespan

Our continuous monitoring of individual cells established that the cell size continuously increases over time, regardless of a cell's ROS level and current size, until the cell bursts (Supplementary Figs. 17-20). It also established the typical (average) sizes of newborn and dying cells. Thus, we could plot the average cell size as a function of time, starting with the typical size at birth and ending with the typical size at death for both the low [GSH] and high [GSH] populations (Fig. 3d and Supplementary Fig. 21). Additionally, using the probability of a cell dividing as a function its size and doubling time, we could determine how many times an average cell divides in its lifetime (i.e., replicative lifespan). This analysis revealed that increasing the GSH level—and thereby decreasing the average ROS level in a cell—caused an increase in the replicative lifespan (Fig. 3d). This makes sense given our finding that ROS inhibits cell divisions. Moreover, consistent with our finding that GSH promotes cell division for cells of every size, the above analysis revealed that the typical cell of the high [GSH] population continued to divide until the last moments of its life as a large cell (Fig. 3d). In contrast, the typical cell of the low [GSH] population could only divide while it is small, during the first half of its 28-day life (Fig. 3d). Finally, the same analysis revealed that decreasing the ROS level causes the cell to increase its size more rapidly, such that that a cell with less ROS typically reached a burst-prone (large) size faster—and thus had a shorter lifespan—than a cell with more ROS (Fig. 3d and Supplementary Fig. 22).

## ROS elongates G1 duration and inhibits G1-to-S transition

We have now established that ROS inhibits cell divisions while allowing for the cell size to continuously increase. Cell division pertains to the S-G2-M (replicative) phase of the cell cycle and cell-size growth pertains to the G1 (growth) phase of the cell cycle[23–25]. Hence, examining how ROS affects the durations of each cell-cycle phase may provide a mechanistic understanding of how ROS prevents cell divisions. To that end, we engineered a yeast strain by fusing GFP to histone H2B, and mCherry to a regulator of G1-to-S transition, Whi5. In this strain, a cell's

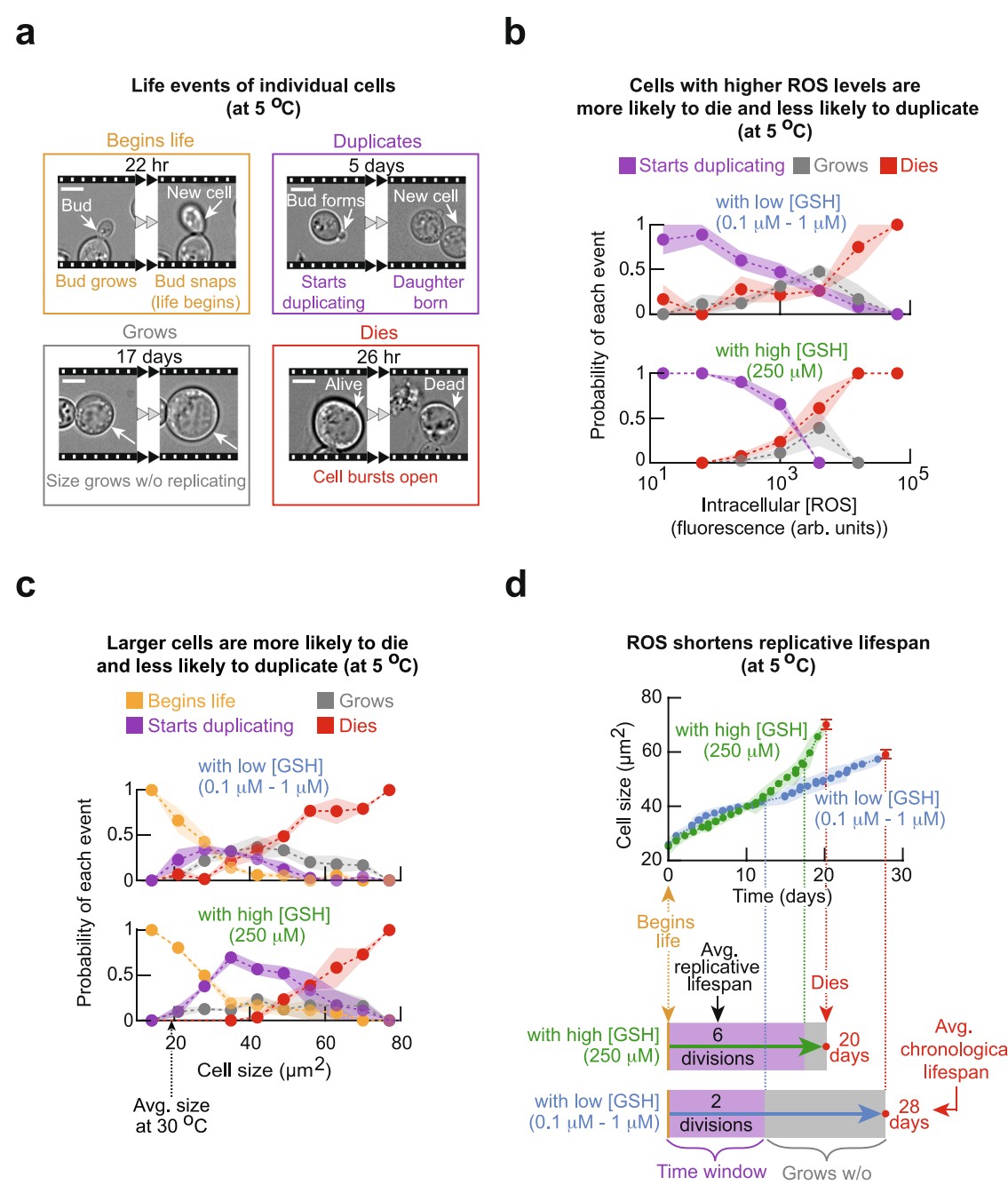

**Fig. 3 | Single-cell analysis of cell-size growth, cell division, and cell death as a function of ROS and cell size. a** Snapshots from time-lapse movies of single cells at 5.0 °C (initially ~8000 cells per ml). Scale bar: 5 μm. To remove transient effects, populations were incubated for 2 weeks at 5.0 °C before the start of the movies on which we based **b** and **c**. **b** Probability of each event (shown in **a**) occurring as a function of intracellular ROS (mitochondrial superoxide) level. Upper half: "with low [GSH]" population is a growing, high-density population that was not given any additional GSH (initially ~6250 cells per ml). Lower half: "with high [GSH]" population is a growing, low-density population that was incubated with 250 μM of GSH (initially ~250 cells per ml). Dots show the mean, and shaded areas show s.e.m. from n = 3 replicate populations. **c** Probability of each event (shown in **a**) occurring as a function of current cell size. Upper half: "with low [GSH]" population as defined in **b** (initially ~ 8000 cells per ml). Lower half: "with high [GSH]" population as defined in **b** (initially ~420 cells per ml). Dots show the mean, and shaded areas show s.e.m. from n = 3 replicate populations. **d** Curves show the size of an average cell over time, from birth to death, at 5.0 °C. Blue curve: cell of "with low [GSH]" population (averaged from n = 330 cells). Green curve: cell of "with high [GSH]" population (averaged from n = 175 cells). Red dot: death occurs. Shaded area and error bars represent s.e.m. of n = 3 replicate populations. Details of how these curves and timelines were constructed are in Supplementary Figs. 19–22. All data in this figure are in the Source Data file.

GFP level was a proxy for its DNA abundance. The mCherry fluorescence migrating from nucleus to cytoplasm marked a cell exiting G1 to enter S phase[20,26,27] (Supplementary Fig. 23). At 5 °C, for one month, we used an epifluorescence microscope to monitor individual cells of the low [GSH] and high [GSH] populations. In both populations, cells that

duplicated typically took one day to continuously replicate their DNA (i.e., GFP level continuously increased) after Whi5 exited their nuclei (Fig. 4a and Supplementary Movie 3). In both populations, cells that did not duplicate continuously increased in size while their DNA level remained constant and their Whi5 remained in their nuclei (Fig. 4b and

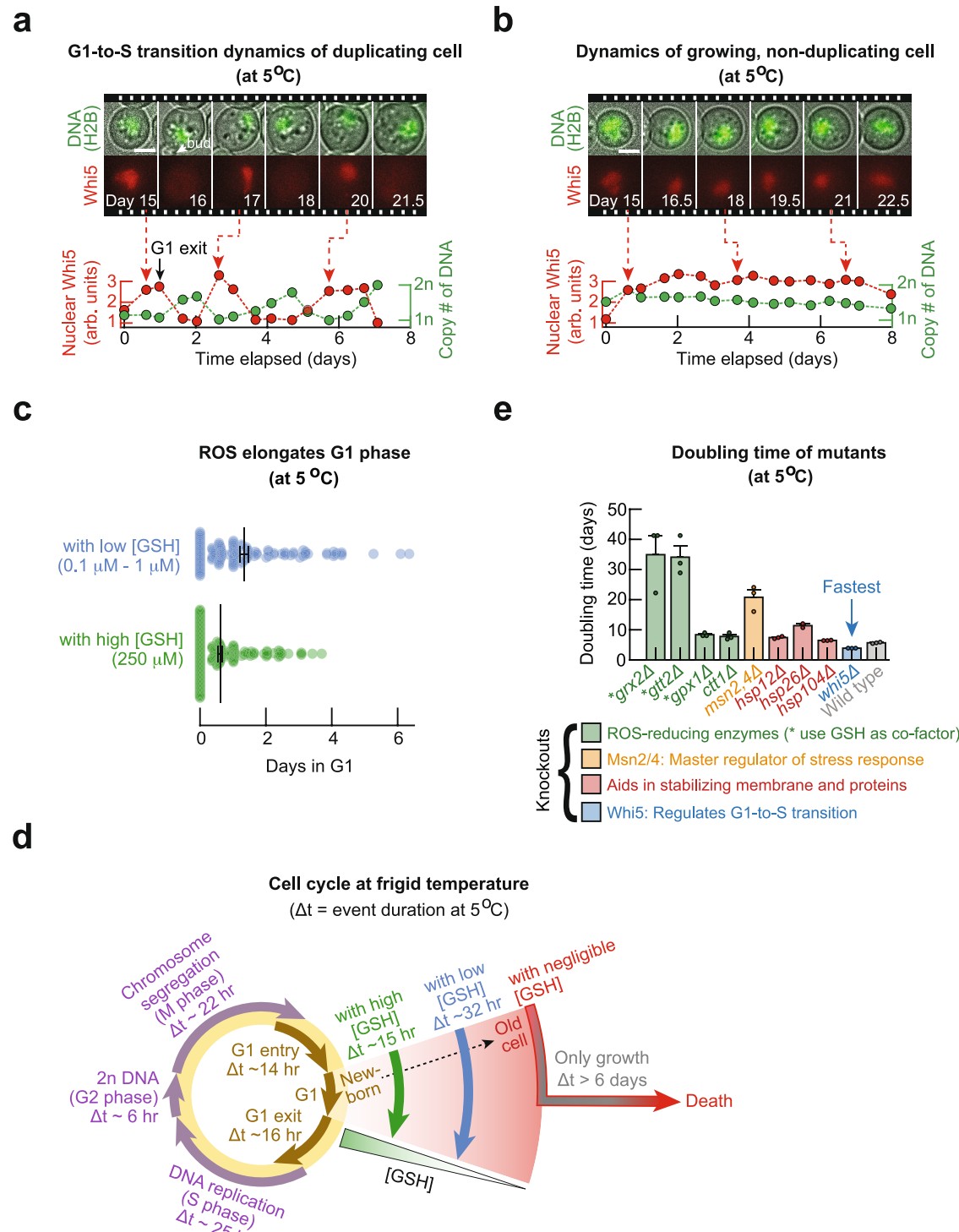

**Fig. 4 | Ultraslow cell-cycle durations tuned with ROS and mutations at frigid temperatures. a, b** Cells were incubated for 2 weeks at 5.0 °C before the start of the time-lapse movies to remove any transient effects. Duplicating cell in **a** and a non-duplicating cell in **b**. Movie strips show composite of brightfield image with fluorescence in GFP (top row, H2B-GFP) or mCherry (bottom row, Whi5-mCherry). Graphs: normalized nuclear Whi5 fluorescence (red) and the copy number of DNA (green). Scale bars are 3 μm. **c** Distribution of G1 duration among cells that divided. Each dot represents one cell. Blue: "with low [GSH]" population (initially ~6250 cells per ml) is a high-density population that grew without any additional GSH (black: average of 32.0 ± 3.0 h). Green: "with high [GSH]' population (initially ~1250 cells

per ml) is a low-density population that grew while incubated with 250 μM of GSH (black: average of 14.7 ± 1.5 h). Black data show mean ± s.e.m.; $n = 3$ biological replicate populations. **d** Average duration of each cell-cycle phase at 5 °C determined by averaging durations from individual cells. G1 duration increases as cell chronologically ages and ROS level increases (GSH level decreases) with measured examples shown (green, blue, red arrows). Purple arrows show replicate phase: S-G2-M. **e** Population doubling time for each knockout strain (all initially ~6250 cells per ml). Gene knocked out is listed and color coded in terms of its function as indicated. Bars show mean with s.e.m. from $n = 3$ biological replicate populations. All data in this figure are in the Source Data file.

Supplementary Movie 4). Hence, for both populations, the non-duplicating cells were stuck in G1.

By monitoring many single cells for a month, we determined the average duration of every major cell-cycle event and how ROS affected each duration at 5 °C (Fig. 4c, d). We found that increasing the GSH level—and thereby decreasing the average ROS level in a cell—caused more cells to exit G1 (Supplementary Figs. 24, 25) and shortened the G1 duration of duplicating cells, from an average of 32 h to 15 h (Fig. 4c, d). Thus, at 5 °C, cells with more ROS spent more time in G1 (growth) phase and were less likely to divide (i.e., less likely to enter S phase). This explains why a cell with an above-threshold concentration of ROS is virtually guaranteed to die by bursting: such a cell cannot leave G1 and thus it increases in size until growing further is physically impossible. We found that nearly every cell that spent more than 6 days in G1 failed to divide and died by bursting (Fig. 4d—red and gray arrows). This suggests that—and as we will show—a cell must exit G1 within a certain time window to complete the cell cycle and that this window is set by the threshold concentration of ROS.

We also found that ROS increases the G1 duration in such a way that a cell typically completed its current cell cycle more slowly than its previous cell cycle (Supplementary Fig. 26). Specifically, a newborn cell typically spent less than 12 h in G1 (Fig. 4d—green arrow) whereas older cells could stay in G1 for more than 12 days until they burst (Fig. 4d—red arrow).

## ROS does not affect G2-S-M (replicative) duration

Reconstructing the cell cycle also revealed that ROS does not affect the durations of G2, S, and M phases. Indeed, for both populations that differed in their ROS levels, a duplicating cell at 5 °C typically took ~25 h to replicate its chromosomes, ~6 h for G2 phase, and ~22 h for mitosis and cytokinesis combined (Fig. 4d - purple arrows; Supplementary Fig. 25). Hence, a duplicating cell typically took a total of ~2 days for the combined S-G2-M (replicative) phase regardless of its ROS level at 5 °C (Fig. 4d - left half of the circle). This result also shows that the wide variation among cells in their doubling times, which we previously noted (Fig. 1c), is due to the variation of their G1 duration but not of their S-G2-M duration.

## Mutations affect cell duplications via G1-to-S transition

We have established that ROS inhibits the G1-to-S transition and thereby prevents cells from dividing while letting their size continue to increase until the cell bursts. Since ROS decreases a cell's chance of exiting G1, we reasoned that eliminating genes for ROS-reducing enzymes that use GSH as a co-factor, such as *GRX2* or *GTT2*[8, 9], would either elongate the G1 phase or cause more cells to die. Both effects would manifest as an increase in a population's doubling time. Indeed, we found that knocking out either one of the two enzymes (*grx2Δ* and *gtt2Δ*) greatly increased the doubling time to 35 days for a high-density population whose doubling time is 6 days when both genes are present (Fig. 4e). Importantly, by constructing several other gene-knockout strains such as knockouts of Msn2 and Msn4[28], we found that the most severe increases in a population's doubling time was caused by knocking out the two ROS-reducing enzymes (*grx2Δ* and *gtt2Δ*) that required GSH as a co-factor (Fig. 4e and Supplementary Figs. 27–29). Hence, disrupting a cell's ROS-reducing ability severely inhibits its ability to duplicate at 5 °C.

Another way to promote cell duplications may be enabling cells with abundant ROS to exit G1. Indeed, we found that knocking out Whi5, an important regulator of G1-to-S transition, caused more cells to duplicate without changing their ROS levels at 5 °C (Supplementary Fig. 30). In fact, the only mutant that we found to have a shorter average doubling time than the wild-type cells at 5 °C was a knockout of Whi5 (*whi5Δ*) (Fig. 4e). These results also held at 1 °C: by using a microscope to continuously monitor individual *whi5Δ* cells for 2 months at 1 °C, we found that knocking out Whi5 increased a cell's

chance of duplicating by 2.5-fold. Specifically, 26% of *whi5Δ* cells could duplicate whereas 11% of wild-type cells of the same population density could duplicate at 1 °C (Fig. 5a; Supplementary Movie 5). At 1 °C, the duplicating cells took ~28 days to complete their S-G2-M phases and G1 duration was typically longer than 51 days (Fig. 5a and Supplementary Figs. 31–33).

## Speed limits for cell cycle at frigid temperatures

We established that a longer doubling time arises from a longer G1 duration which, in turn, arises from having more ROS. Once the ROS level increases to an above-threshold concentration, a cell has virtually no chance of completing the cell cycle and is guaranteed to burst without dividing. This reasoning establishes that there must be a longest allowed doubling time for each temperature. It also follows that there must be a shortest allowed doubling time for each temperature because the S-G2-M phase has a fixed duration that is independent of ROS. In the remainder of our paper, we focus on combining all our findings thus far to determine the possible pace at which a yeast cell can cycle and the speed limits for this pace—the longest and shortest allowed doubling times—at every temperature above 0 °C. We reasoned that global gene-expression machineries (e.g., ribosomes, RNA polymerases) would affect a cell's doubling time[29, 30] because we found that ROS-reducing enzymes, which require gene expression to produce, are critical for cell duplications. Moreover, we reasoned that a cell needs time to build a daughter cell and replace key cellular components that ROS damaged. These processes involve the same, global gene-expression machineries. Hence, we hypothesized that the combined working speed of global gene-expression machineries are major factors in determining the doubling time and sought to measure the genome-wide transcription rate and protein-synthesis rate at frigid temperatures.

## Genome-wide transcription rate at frigid temperatures

To determine a genome-wide transcription rate in yeast at frigid temperatures, we incubated the low [GSH] population at various temperatures (1 °C, 5 °C, or 30 °C) in a growth medium containing the nucleotide analog, 4-thiouracil (4tU). 4tU incorporates into every newly synthesized RNA. We quantified the abundance of all newly made transcripts at different time points by using next-generation sequencing of the 4tU-labeled mRNA (4tU-seq, see Methods and Supplementary Figs. 34, 35). Additionally, we used single-molecule RNA FISH[31, 32] on endogenous yeast genes to measure the integer copy-number of transcripts, at their steady-state expression levels, for each temperature. We then converted the abundance of mRNA from the 4tU-seq to an integer copy-number of mRNA per cell (Supplementary Fig. 36). This revealed a genome-wide transcription dynamics at each temperature (as # of mRNA per cell per hour) (Fig. 5b). A rate equation[33, 34] with a constitutive synthesis and degradation of mRNA recapitulated the data (Fig. 5b and Supplementary Notes). This fitting revealed that, on average, a cell synthesized 700 mRNA molecules per hour at 1 °C, 2000 mRNA molecules per hour at 5 °C, and 58,000 mRNA molecules per hour at 30 °C. Moreover, the fitting revealed that the average half-life of mRNA at each temperature: 14 h at 1 °C, 7 h at 5 °C, and 20 min at 30 °C (Supplementary Figs. 37–40). These measurements established that transcription occurs on the order of hours to days at frigid temperatures.

## Protein-synthesis rate at frigid temperatures

To determine the timescale for making a functional protein, starting from transcription initiation, we constructed a strain in which galactose induced an expression of mCherry. After adding galactose to the growth medium of the low [GSH] population, we measured mCherry protein abundance in individual cells for up to 2 weeks at various temperatures (Fig. 5c; Supplementary Movie 6). The global machineries for gene expression, such as RNA polymerases and ribosomes,

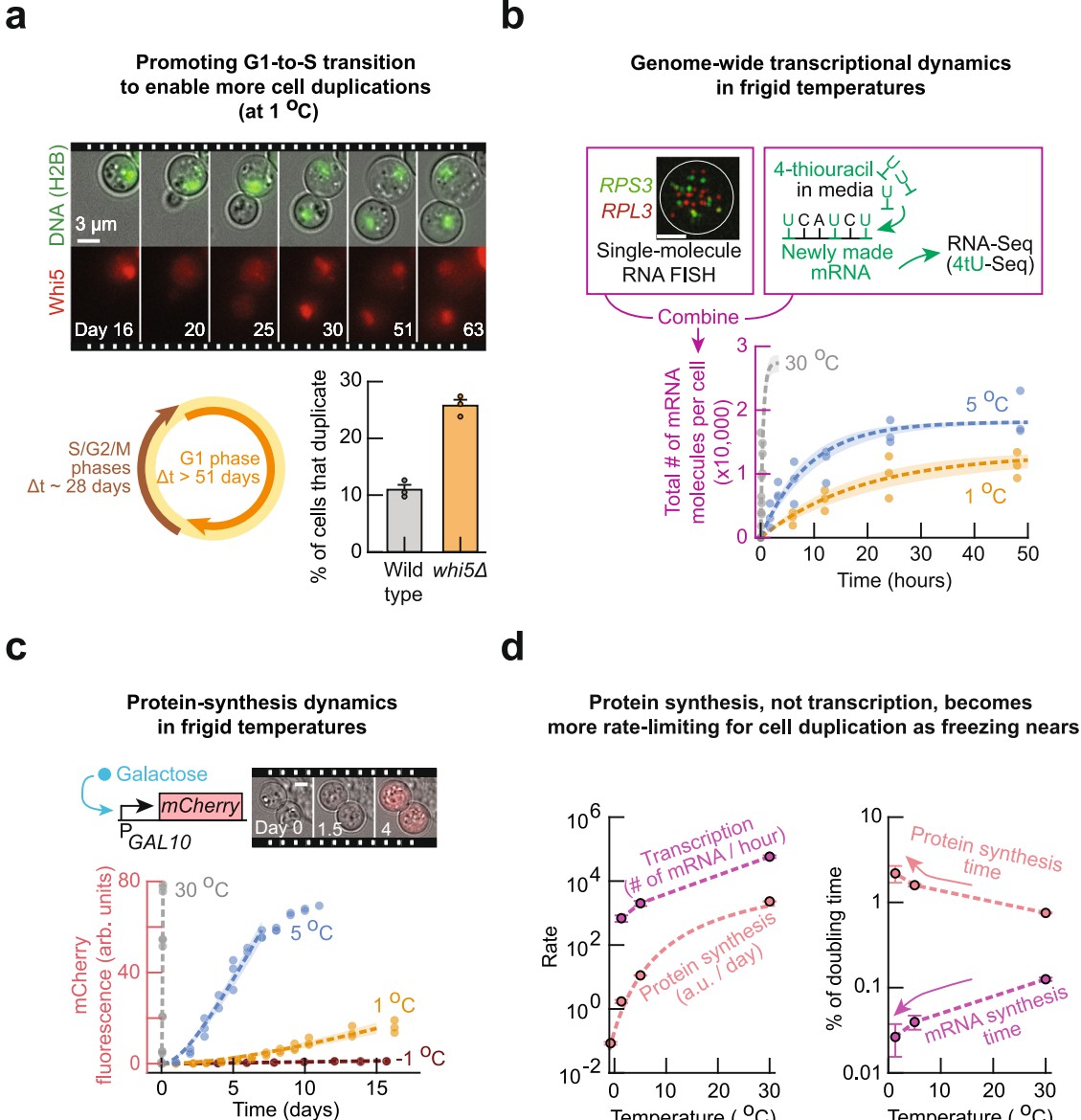

**Fig. 5 | Genome-wide transcription rate and protein-synthesis rate at frigid temperatures. a** Time-lapse movie of a duplicating cell at 1.0 °C. Composite of brightfield image and H2B-GFP (top) or Whi5-mCherry (bottom). Scale bar is 3 μm. Circle shows average duration of cell-cycle phases at 1.0 °C: 28.4 ± 3.2 days for S-G2-M phase (brown arrow), and at least 51 days for G1 (equal to duration of the time-lapse, orange arrow). Bars represent mean ± s.e.m.; *n* = 3 biological replicate populations. **b** Measuring genome-wide transcriptional dynamics by combining metabolically labeled RNA-sequencing with single-molecule RNA FISH (details in Methods). Graph: total number of newly synthesized mRNA over time at 30.0 °C (gray), 5.0 °C (blue) and 1.0 °C (orange). Image shows mRNAs in a single cell at 1.0 °C as visualized by single-molecule RNA FISH, with a composite of *RPS3* (green) and *RPL3* (red). White circle shows an outline of a cell. Scale bar is 2 μm. **c** Measuring protein-synthesis dynamics in individual cells. Movie strip: mCherry expression at

5.0 °C after 0, 1.5, and 4 days of incubation with 2% galactose. Images are composite of brightfield and mCherry fluorescence. Scale bar is 5 μm. In **b**, **c**, dots show raw measurements, dotted lines show a model fitted (Supplementary Notes), and shaded areas show s.e.m. of fitted parameters. *n* = 3 biological replicates. **d** Plot of rates extracted from **b** and **c**: Left graph: transcription rate (# of mRNA per hour) and protein synthesis rate ([mCherry] arb. units per day). Red dotted line shows fit of Arrhenius-type equation to protein synthesis rate (Supplementary Notes). Right graph: characteristic time for protein synthesis and transcription, both are relative to the average doubling time at each temperature. Characteristic time is the time to synthesize 100 arb. units of mCherry or 10,000 mRNAs. In both graphs, data shown as mean ± s.e.m.; *n* = 3 biological replicates. All data in this figure are in the Source Data file.

must function at least as rapidly as the time taken for mCherry fluorescence to increase in cells. A rate equation for gene induction recapitulated our data at all temperatures to yield a mCherry-synthesis rate at each temperature (Fig. 5c and Supplementary Figs. 41, 42; details in Supplementary Notes). By normalizing this rate for each temperature to that of 30 °C, we obtained a fold-reduction in the mCherry-synthesis rate (protein-synthesis rate) which represents a slowing down of the global gene-expression machineries (e.g., ribosomes, RNA polymerases). In support of our result, we found that an Arrhenius-type

function recapitulated how the protein-synthesis rate decreased as temperature decreased towards 0 °C (Fig. 5d and Supplementary Notes). These measurements established that protein synthesis, starting from transcription initiation, occurred on the order of a week or longer at frigid temperatures.

**Protein synthesis becomes more limiting for cell duplication**

The above measurements of transcription and protein-synthesis rates were performed with the low [GSH] population. By measuring the rates

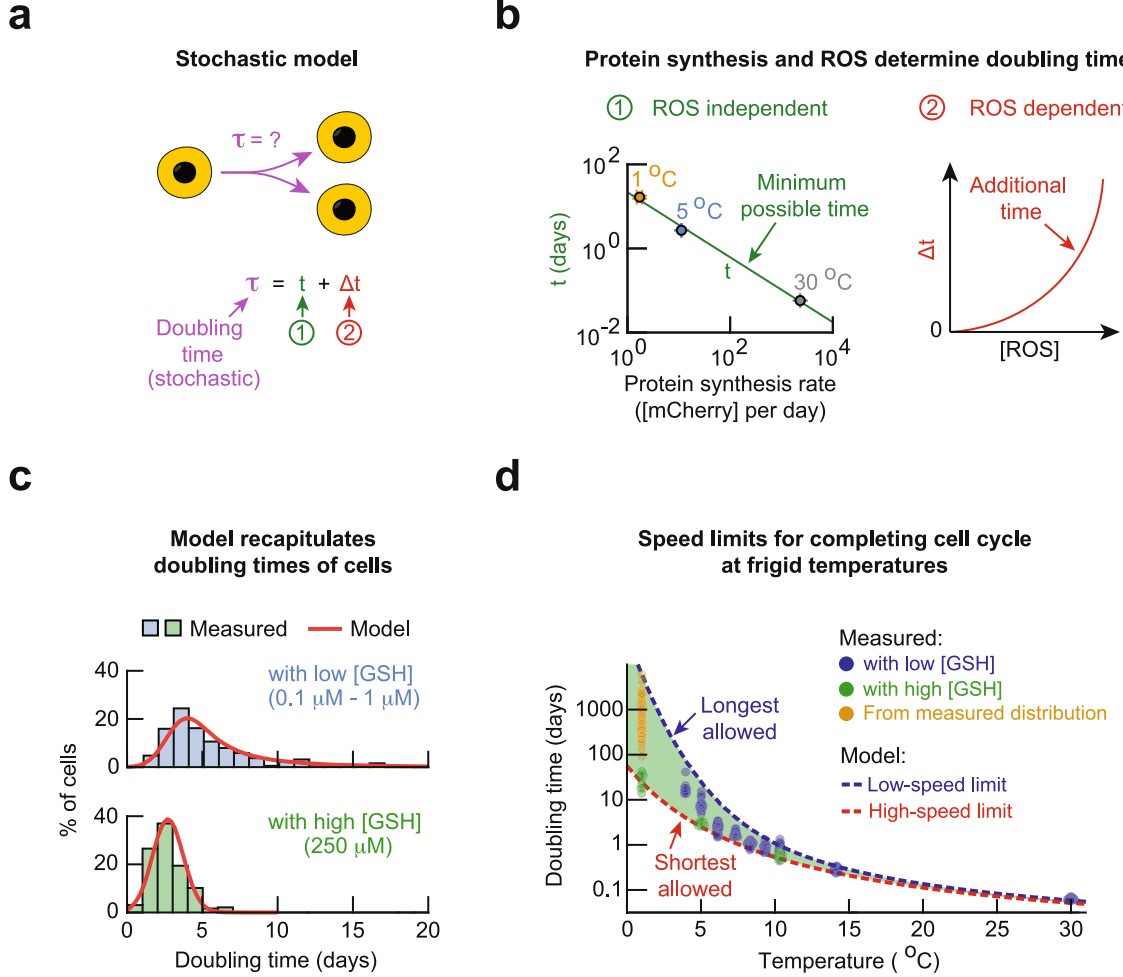

**Fig. 6 | Mathematical model explains origin of low-speed and high-speed limits for completing cell cycle at frigid temperatures. a** Description of the stochastic model (Supplementary Notes). A cell's doubling time τ is a stochastic variable that is dictated by a ROS-independent minimum duration *t* (labeled as "1") and a stochastic, ROS-dependent duration Δ*t* (labeled as "2"). **b** Minimum possible doubling time *t* (taken from populations with 250 μM GSH at 5 °C or the average duration of S-G2-M at 1 °C) as a function of the protein-synthesis rate. Error bars show s.e.m. of *n* = 3 biological replicates. Solid green line shows a power-law fit (exponent of 0.77 ± 0.05, Pearson correlation-coefficient β = 0.9979). The additional duration Δ*t* is determined by the intracellular ROS level and varies among cells (red line). **c** Model recapitulates the doubling times of cells at 5.0 °C with and without added

GSH. Histograms are measured values. Red curves are predictions of the model. **d** Model produces longest allowed doubling time (blue curve) and shorted allowed doubling time (red curve) for each temperature. The longest allowed doubling times are the doubling times of cells with the threshold concentration of ROS. Model predicts that a cell's doubling time must fall within the green shaded region. Experimental data are from "with low [GSH]" populations as defined previously (blue dots), "with high [GSH]" populations (green dots, >5 °C), and 1 °C (green dots). Yellow dots are doubling times of individual cells that we extrapolated from measured distribution of cell-cycle times at 1 °C (Supplementary Notes). All data in this figure are in the Source Data file.

again but now in the high [GSH] population, we discovered that ROS did not affect either of the rates at any temperature (Supplementary Figs. 43–45). Thus, we can make general conclusions about the transcription and protein-synthesis rates at frigid temperatures by examining the results for the low [GSH] population. By comparing the transcriptional rate with the protein-synthesis rate at various temperatures for the low [GSH] population, we found that the protein-synthesis rate decreased faster compared to the genome-wide transcription rate as temperature decreased toward 0 °C (Fig. 5d and Supplementary Fig. 46). This was more evident when we compared how the inverse of each rate changed as a percentage of the doubling time at each temperature. The inverse of each rate yields a characteristic time for a given process (transcription or protein synthesis). Plotting the characteristic time as a fraction of the doubling time at each temperature reveals that as the temperature decreases, mRNA-synthesis time becomes more negligible whereas protein-synthesis time becomes a larger fraction of the doubling time (Fig. 5d). Since the characteristic time for protein synthesis includes the characteristic time for transcription, this result

establishes that, as temperature approaches 0 °C, the working speeds of translational and post-translational machineries become more rate limiting for cell duplication whereas transcription becomes less rate limiting for cell duplication.

## Mathematical model explains speed limits for cell cycle

We can now determine how a cell's doubling time arises from its ROS level and protein-synthesis rate at each temperature. We established that cells with more ROS have longer doubling times and that they rely on ROS-reducing enzymes. Hence, a cell with a minimal ROS would have the shortest doubling time *t* which would depend only on the protein-synthesis timescale which is not affected by ROS. For a cell with a non-negligible level of ROS, it would need an additional time Δ*t*, on top of *t*, to remove ROS. Cells with higher ROS would have a larger Δ*t*. Thus, a cell's doubling time τ is (Fig. 6a):

$$\tau = t + \triangle t \qquad (1)$$

Like the protein-synthesis rate, we found that an Arrhenius-type function recapitulated the measured doubling times of cells that had the least amounts of ROS. Consequently, we discovered a power-law relationship[35, 36] between $t$ and the average protein-synthesis rate $r$: $t \sim r^{-0.77}$ (Fig. 6b). To obtain $\Delta t$, which varies among cells as a stochastic variable, we used the fact that the measured distribution of ROS levels among cells was approximately log-normal and that a cell needs time to build ROS-reducing enzymes which depends on the protein-synthesis rate. These two considerations lead to $\triangle t \sim \frac{1}{r} \cdot \exp([ROS])$ (Fig. 6b and Supplementary Notes). With the $t$ and $\Delta t$ as defined above, we performed stochastic simulations in which we used the observed threshold-concentration of ROS to determine whether a cell divides or not and, for a dividing cell, its doubling time (details in Supplementary Notes). The simulation reproduced the measured distribution of doubling times among cells (shown for 5 °C in Fig. 6c; Supplementary Fig. 47). Importantly, when we plotted together all the measured doubling times for every temperature, we found that all data points lay between the shortest and longest allowed doubling times that the model dictated at every temperature (Fig. 6d). Hence, the data validate our model's explanation of the origin of speed limits for completing the cell cycle. Specifically, the shortest allowed doubling time $t$, which corresponds to a high-speed limit, is defined by the protein-synthesis rate: $t \sim r^{-0.77}$. The model shows that cells cannot complete the cell cycle any faster than the temperature-specific, high-speed limit because cells cannot synthesize proteins at an arbitrarily fast pace. The longest allowed doubling time $\tau_{max}$, which corresponds to a low-speed limit, is determined by the protein-synthesis rate and the threshold concentration of ROS ($[ROS]_{\text{threshold}}$):

$$\tau_{\text{max}} \propto r^{-0.77} + \frac{1}{r} \cdot \exp([ROS]_{\text{threshold}}) \qquad (2)$$

The model shows that cells cannot complete the cell cycle any slower than the temperature-specific, low-speed limit. Near 0 °C, the low-speed limit corresponds to an extremely long doubling time (e.g., ~3 years at 1 °C). While such cell duplications may be possible, their likelihood in the model are so low that observing such a cell duplication is extremely unlikely (Supplementary Fig. 31).

## Discussion

Without knowing any details, one can state that a cell cannot duplicate at an arbitrarily fast pace for any temperature. But it is not obvious that a cell cannot take an arbitrarily long time to complete the cell cycle. Here, we discovered high- and low-speed limits at each, frigid temperature and how they arise from an interplay between the protein-synthesis rate and ROS. Crucially, we found no evidence of an upper bound for the doubling time: yeast's life can be slowed down to an arbitrarily slow pace by bringing the temperature arbitrarily close to the freezing point. However, we also found that self-replication, while possible, becomes exceedingly unlikely as temperature approaches the growth medium's freezing point.

We uncovered a temperature-dependent growth law for yeast by examining the temperature-dependent quantitative relationships among the rates of genome-wide transcription, protein synthesis, and cell proliferation (Supplementary Discussion). These relationships have been unclear despite previous studies having examined how temperature affects specific genes in microbes[6, 37–40]. We discovered how slowly gene-expression machineries can function and how their slow functioning constrains the cell-cycling pace. This work thus revealed quantitative limits to thermally slowing down a cell's self-replicating dynamics.

## Methods

### Yeast strains

The wild-type, haploid yeast strain W303 that we used is from Euroscarf with the official strain name, 20,000 A. It is isogenic to another laboratory-standard haploid yeast strain, W303a, and has the following genotype: *MATa; his3-11_15; leu2-3_112; ura3-1; trp1Δ2; ade2-1; can1-100*.

### Growth media

We cultured all yeasts in defined, minimal media (SC) that consisted of (all from Formedium): Yeast Nitrogen Base (YNB) media (cat. No. CYN0410), Complete Supplement Mixture (CSM, cat. No DCS0019) containing all the essential amino acids and vitamins, and glucose at a saturating concentration (2% = 2 g per 100 ml, Melford Biolaboratories Ltd., cat. No. G32040). The agar pads, which we used for growing yeast colonies, contained 2%-agar (VWRChemicals), Yeast Extract and Peptone (YEP) (Melford Biolaboratories Ltd., cat. No. Y20020 and P20240 respectively), and 2% (w/v) glucose.

### Flow cytometry

We used BD FACSCelesta with a High-Throughput Sampler and lasers with the following wave lengths: 405 nm (violet), 488 nm (blue) and 561 nm (yellow/green). Data were gathered with BD FACSDiVa 8.0 software. We calibrated the FSC and SSC gates to detect only yeast cells (FSC-PMT = 681 V, SSC-PMT = 264 V, GFP-PMT = 485 V, mCherry-PMT = 498 V. As a control, flowing dPBS yielded no detected events). The number of cells per mL that we plotted in our growth experiments is proportional to the number of events (yeast cells) that the flow cytometer measured in an aliquot of cells with a defined volume. We measured the GFP fluorescence with a FIT-C channel and the mCherry fluorescence with a mCherry channel. We analyzed the flow cytometer data with a custom MATLAB script (MathWorks; R2020).

### Growth experiments

In a typical growth experiment, we first picked a single yeast colony from an agar plate and then incubated it at 30 °C for ~14 h in 5 ml of minimal medium. Afterward, we took a 20 μl aliquot from the 5 ml culture, diluted it to a known volume and then flowed it through our flow cytometer to determine the 5 ml culture's population-density (# of cells per ml). We then serially diluted the culture into fresh minimal media to a desired initial population-density for a growth experiment at the desired temperature. Specifically, we distributed 5 ml of diluted cells to individual wells in a "brick" with twenty-four 10 ml wells (Whatman, 24-well × 10 ml assay collection and analysis microplate). This ensured that we had four identical replicate cultures for each initial population-density. We sealed each brick with a breathable film (Diversified Biotech: Breathe-Easy), covered it with a custom-made Styrofoam-cap for insulation, and incubated it in a compressor-cooled, high-precision thermostatic incubators (Memmert ICP260) that stably maintained their target temperature throughout the course of our growth-experiments, with a typical standard deviation of 0.052 °C over time (deviation measured over several days—see Supplementary Fig. 1). Throughout the incubation, the cultures in the brick were constantly shaken at 400 rpm on a plate shaker (Eppendorf MixMate) that we kept in the incubator. To measure their population densities, we took a small aliquot (typically 250 μl) from each well, diluted it with dPBS (Fisher Bioreagents) into a 96-well plate (Sarstedt, Cat. #9020411), and then flowed it through the flow cytometer which gave us the # of cells per ml. Alternatively, the cultures were distributed into glass tubes, that were kept in the incubator and constantly mixed using a rotator set to 40 rpm.

### Measuring the percentage of dead cells

Cells were incubated as described in the Methods section titled "growth experiments". After ~2 weeks of incubation, we took aliquots of each culture at each time point, and then stained the cells for 20 min

with 1 µg per ml of propidium iodide (Thermo Fisher Scientific, cat. No. P3566). We then flowed these (stained) cells through our flow cytometer and measured the number of cells that were unstained by propidium iodide—these cells have intact membranes, and are assumed to be alive (conversely, stained cells have lost membrane integrity and are assumed dead[41]. We then used the total number of cells and the number of dead or alive cells to extract the growth rate and death rate during several weeks. For this we assumed a simple stochastic growth model that we fitted to our data (see Supplementary Notes).

### Microscope sample preparation

All microscopy imaging was performed with 96-well glass-bottom imaging plates (cat. No. 5221-20, Zell-Kontact). Before each sample preparation, the glass bottom of the wells was pre-treated with 0.1 mM concanavalin A for 20 min at room temperature (ConA, Cat. No. C2010, Sigma-Aldrich). We then removed the conA and added an appropriate number of cells to each well. Typically, we added an aliquot containing ~7500 cells and supplemented with sufficient dPBS (Gibco, Life Technologies Limited, cat. No. 14190-144) such that the volume of each well was 200 µl. The plate was then centrifuged at $107 \times g$ for 5 min using a centrifuge (Eppendorf, 5810R) that was precooled at the desired temperature (e.g., 5 °C).

### Microscope data acquisition and time-lapse

We used an Olympus IX81 inverted, epifluorescence, wide-field microscope. Temperature was kept constant during imaging by an incubator cage (OKO Lab) that enclosed the microscope. Fluorescent proteins or fluorescent probes were excited using a wide-spectrum lamp (AMH-600-F6S, Andor) and images were acquired with an EM-CCD Luca R camera (Andor) and IQ3 software v3.2 (Andor). For time-lapse movies, we prepared yeast cells as described in the Methods section titled, "growth experiments", for 2 weeks. Aliquots were then transferred to 96-well imaging plates as described in the Methods section titled, "microscope sample preparation", except that the cultures were not diluted with dPBS. The imaging plates that contained yeast samples were subsequently incubated at the desired temperature throughout the time-lapse (e.g., 5 °C for typically ~3 weeks), and transported and kept on ice for imaging (typically once every day). We checked that the transport and microscopy—usually less than 15 min during which the imaging plate and samples inevitably warm up—had no observable influence on the samples. To do this, we compared two imaging plates that contained aliquots of the same samples. One plate was imaged once every day for 3 weeks as described above. The other plate was only imaged once, after the 3 weeks of incubation and was never transported on ice or warmed up. We found no difference between the cultures in the two plates—in terms of cell density, ROS stress or cell size—after 3 weeks of incubation.

### Microscope data analysis

We processed the microscope data using ImageJ (1.53c) and MATLAB (R2020). Specifically, we segmented the cells by creating oval masks containing the cells and extracted fluorescence values or cell size from the area inside these masks. We computed the fluorescence for each cell by taking the maximum intensity of 20 images spaced 0.2 µm apart in a z-stack. Finally, we corrected for the background fluorescence by subtracting the average (maximum) background fluorescence in the field-of view from the value obtained for each cell.

### Measuring extracellular reduced and oxidized glutathione

To quantify extracellular glutathione, we isolated the growth media from liquid cultures by flowing liquid cultures through a 0.2 µm pore filter (VWR). We ensured that no cells remained in the filtered media by flowing an aliquot through our flow cytometer. We then measured the total concentration glutathione in the filtered media as described in

the manufacturers' protocol (quantification kit for oxidized and reduced glutathione (cat. No, 38185, sigma aldrich)). To quantify both the concentration of oxidized and reduced glutathione, we took two 200 µl aliquots of the filtered media. To one of the aliquots, we then added 4 µl of masking agent provided with the kit (most likely 2-vinylpyridine in ethanol at a final concentration of ~3 mM). All aliquots were then incubated for 1 h at 37 °C together with standard curves for reduced glutathione (Cat. No. G4251, sigma Aldrich) without masking agent and oxidized glutathione (G4376, Sigma Aldrich) with masking agent. (During incubation, the masking agent irreversibly binds and thereby removes reduced glutathione. The assay subsequently only detects oxidized glutathione in the samples. We verified that this protocol indeed quantifies the amount of oxidized and reduced glutathione. We found that the masking agent removed ~90% of the reduced glutathione during incubation, resulting in a false signal of ~10% reduced glutathione.). We used a spectrophotometer (Spectrostar nano, BMG labtech) to measure the optical absorbance at 415 nm.

### Measuring the concentration of intracellular ROS

We prepared yeast cells as described in the Methods section titled, "growth experiments", and incubated the cultures for 2 weeks. We then transferred aliquots to 96-well imaging plates as described in the "Methods" section titled, "Microscope sample preparation", except that the cultures were not diluted with dPBS. We next removed the supernatant and washed the cells twice with precooled dPBS at 5 °C to remove thiols from the growth media (e.g., cysteine). We then added an indicator dye and incubated the cells for 30 min at 5 °C. As indicator dye, we used the dye named "mitoSOX red" to stain intracellular superoxide (at 5 µM final concentration, Thermo Fisher scientific Cat. No. M36008) or a dye named "cellROX orange" to stain intracellular ROS (at 5 µM final concentration, Thermo Fisher scientific Cat. No. C10443). Finally, we removed the excess dye by washing the cells twice with dPBS and imaged the cells with a microscope as described in the "Methods" section titled, "microscope data acquisition and time-lapse". For a co-stain of multiple indicator dyes, we stained the cells as described above by co-incubating the cells with multiple dyes at the same time. For time-lapse movies of cells stained with the indicator dyes, we followed the above protocol with the following modifications. After preparing the cells and washing away the excess dye, we placed back the supernatant (growth media) that we took away before and kept at 5 °C when incubating the cells with the dye. We then proceeded with the microscopy time-lapse as described in the "Methods" section titled, "microscope data acquisition and time-lapse".

### Measuring extracellular ROS production rate

To measure the extracellular oxidation rate, we used a probe called dihydroethidium (DHE, Thermo Fischer Scientific, Cat. No. D11347) that becomes fluorescent upon oxidation by ROS (superoxide). Dihydroethidium is usually used to detect intracellular ROS, and it intercalates with DNA giving a bright signal in the nucleus. Instead, we used dihydroethidium to detect extracellular ROS where no cellular components were present. To still facilitate the fluorescent signal of dihydroethidium, we added herring sperm DNA (Promega, Cat. D1816) to our samples with a 0.2 mg per ml final concentration. Right before measuring fluorescence, we added dihydroethidium to our samples with a 20 µM final concentration. We then transferred the samples to a flat-bottom 96-well plate (Sarstedt, Cat. #82.1581.001) using 150 µl per well. Directly after, fluorescence was measured using a plate reader (Synergy HTX Multi-Mode Microplate Reader, Biotek) every 3 min for 2 h. Data were gathered with the BioTek Synergy HTX software (version 2018). Fluorescence was measured with excitation at 500 nm and emission at 620 nm. For measuring the oxidation rate at 30 °C, we pre-warmed all media and set our plate reader to incubate the samples at 30 °C during the measurements. For measuring the oxidation rate at

5 °C, we prepared 5 mL of each sample that we precooled at 5 °C. During the experiment, we incubated the samples at 5 °C for a day. We took 150 μl aliquots of each sample every hour and transferred these into a 96-well plate. Directly after we measured the fluorescence of these aliquots with our plate reader. For all samples, we measured and averaged the fluorescence of three technical replicates. All measurements included controls consisting of pure water or regular SD media. The oxidation rate was determined by determining the slope (arb. units per second) of the fluorescence curve during ~1 h (at 30 °C), starting typically after measuring 10 min with the plate reader. We used the following scavengers of ROS (from Sigma Aldrich): tiron (4,5-dihydroxy-1,3-benzenedisulfonic acid disodium salt monohydrate, cat. No. 172553), trolox ((±)−6-Hydroxy-2,5,7,8-tetramethylchromane-2-carboxylic acid, cat. No. 238813). Samples with ROS scavengers were compared to a control in appropriate solvent (for example, a sample having trolox that is dissolved in DMSO was compared to a sample having only DMSO).

### Limited nutrients experiment

To test the effect of nutrients on the growth of cells at low temperatures, we prepared fresh wild-type yeast cells as described in the Methods section titled, "growth experiments", except that we limited the amount of nutrients in the fresh growth media. Specifically, we diluted the minimal media with various amounts of water, and then supplemented each media with 2% glucose. Thus, each media contained a known percentage (0–100%) of the nutrients that are in regular minimal media and 2% of glucose. After transferring fresh populations of cells into each media, we incubated the cultures to 5 °C and measured their population density over time as described in the Methods section titled, "growth experiments".

### Measuring extracellular ROS production during nutrient depletion

To measure the extracellular oxidation rate that cells experience during their incubation at low temperatures, we prepared cultures of our wild-type yeast at various starting densities as described in the Methods section titled, "growth experiments". We then incubated the cells at the 5 °C, and measured the oxidation rate in the growth media over time. To do so, we took aliquots of the cultures that we kept at 5 °C, and flowed them through a 0.2 μm pore filter (VWR, cellulose-acetate membrane). We then directly proceeded to measure the ROS production rate in the supernatant as described in the Methods section titled, "measuring extracellular ROS production rate".

### Measuring the cell cycle progression

We used a strain, termed "cell-cycle marker strain", to measure the progression of the cell cycle at low temperatures. We prepared cultures of the cell-cycle marker strain as described in the Methods section titled, "growth experiments". After 2 weeks, we transferred aliquots of the cultures to a pre-cooled 96-well microscopy plate that we kept on ice. The microscopy plate was further prepared as described in the Methods section titled, "microscope sample preparation". We kept the plate at the desired temperature (e.g., 5 °C) for one day, and then proceeded by taking a snapshot of each sample twice per day for time-lapse movies as described in the Methods section titled, "microscopy data acquisition". We analyzed the time-lapse movies as described in the Methods section titled, "microscope data analysis". In short, to quantify the amount of nuclear Whi5-mCherry and H2B-GFP, we first located the nucleus by segmenting the GFP fluorescence of each cell using a threshold GFP fluorescence that we kept fixed for all cells and time points. The nucleus was then the group of pixels whose fluorescence exceeded this threshold. We then determined the total mCherry and GFP fluorescence within the cell's nucleus. From this fluorescence we subtracted the average background fluorescence in

the field-of-view from the value obtained for each cell. Finally, to obtain the copy number of DNA we rescaled the nuclear GFP between the average minimum and maximum GFP fluorescence that we observed for duplicating cells, and to obtain the amount of nuclear Whi5 we took the ratio of the nuclear and cytoplasmic mCherry (Supplementary Fig. 23).

### Mutant yeasts

We constructed several mutant strains in which we removed genes involved in (oxidative) stress-response. In short, we designed primers whose ends were homologous to the flanking regions of the desired gene to be knocked out. Using these primers, we amplified a selection marker by PCR, and knocked out the desired gene in the wild-type yeast via homologous recombination. Mutants were selected on YPD selection plates and knockouts were verified by PCR. Specifically, we knocked out the genes for the stress-response transcriptional activators (*MSN2*), membrane organization (*HSP12*), disaggregase (*HSP104*) glutathione s-transferase (*GTT2*) and glutathione peroxidase (*GPX1*) using the HygB selection marker and YPD plates containing hygromycin B. We also knocked out genes for a suppression of protein aggregation (HSP26), glutaredoxin (*GRX2*), catalase (*CTT1*) and the transcriptional regulator of G1-to-S transition (*WHI5*) using the NatMX selection marker and YPD plates containing nourseothricin. The *MSN2, MSN4* double knockout was constructed by removing, sequentially, first the *MSN2* gene and then the *MSN4* gene. We thus obtained several mutants that lacked genes for transcriptional regulation (*msn2, msn4*Δ-strain, *whi5*Δ-strain) or that lacked genes for the oxidative stress response (*gtt2*Δ-strain, *gpx1*Δ-strain, *grx2*Δ-strain and *ctt1*Δ-strain).

### FISH probes

We designed single-molecule FISH probes to detect mCherry-mRNA. For this we used the Stellaris FISH probe designer (LGC Biosearch Technologies; www.biosearchtech.com). The set of probes (25 probes) were designed to attach to the full length of mCherry RNA and were coupled to Quasar 670 (a Cy5 analog, LGC Biosearch Technologies). We also designed FISH probes to detect mRNA of endogenous yeast genes to convert TPM values from our 4tU RNA-sequencing data to integer numbers of RNA per cell. For this, we used probes for RPS3 (30 probes coupled to Quasar 670), RPL3 (48 probes coupled to Quasar 570, a Cy3 analog LGC Biosearch Technologies), RPB1 (48 probes coupled to Quasar 670) and RPB3 (40 probes coupled to Quasar 570). The excitation and emmission peaks of these fluorophores are ex. 548/em. 566 nm (Quasar 570) and ex. 647/em. 760 nm (Quasar 670).

### Single-molecule RNA FISH

We used the standard protocol for single-molecule RNA FISH in yeast, as described in "Protocol for *S. cerevisiae* from Stellaris RNA FISH" (LGC Biosearch Technologies). Finally, we made sure to image the fluorescence of Quasar 670 probes first during our measurements, as this was the dye most sensitive to photo bleaching.

### Measuring single-gene transcription rate

We used a strain termed, "mCherry-inducible strain", to measure the transcription rate. We prepared cultures of the mCherry-inducible strain as described in the Methods section titled, "growth experiments", except that we used minimal media (SC) containing 2% raffinose as the growth media. The cultures were incubated in 500 ml Erlenmeyer flasks 5 °C for 2 weeks on an Eppendorf platform shaker set to 125 rpm. After 2 weeks, we supplemented the cultures with 2% galactose. After further incubating the cultures at 5 °C for the desired amounts of time, we transferred aliquots of the cultures (typically ~10 ml) to 15 ml tubes containing 37% formaldehyde such that the final volume formaldehyde was 10%. We then proceeded with RNA FISH as described in the Methods section titled, "single-molecule RNA FISH".

## Measuring single-gene expression rate

We used a strain named, "inducible mCherry strain", to measure the gene expression rate. We prepared cultures of the mCherry-inducible strain as described in the Methods section titled, "growth experiments", except that the minimal media (SC) contained 2% raffinose as the carbon source. We then incubated the cultures for two weeks in glass tubes at 5 °C in a rotator at 40 rpm. After 2 weeks, we transferred aliquots of the cultures to a pre-cooled 96-well microscopy plate that we kept on ice. The microscopy plate was further prepared as described in the Methods section titled, "microscope sample preparation". As media, we used fresh, pre-cooled SC containing 2% raffinose to dilute the aliquots to the desired density on the microscopy plate. We kept the plate at the desired temperature (e.g., 5 °C) for one day, and then took a snapshot of the populations as described in the Methods section titled, "microscope data acquisition and time-lapse". Finally, we supplemented each sample on the microscopy plate with 2% galactose to induce the expression of mCherry. We then proceeded by taking a snapshot of each sample twice per day for time-lapse movies as described in the Methods section titled, "microscopy data acquisition". In parallel, we also added 2% galactose to the original cultures that were kept in the rotator at 5 °C. We then measured the average mCherry fluorescence of the population twice per day by flowing aliquots of the cultures through our flow cytometer as described in the Methods sections titled, "growth experiments" and "flow cytometry".

## Preparing cells and RNA extraction

To prepare fresh cells we first picked a single yeast colony from an agar plate and then incubated it at 30 °C for ~14 h in 6 ml of SD media. We then took aliquots of this culture and spun them down using a centrifuge. For each aliquot, we removed the supernatant and resuspended the pellet in fresh media of the desired composition (for example, fresh SD, or fresh SD containing 250 μM glutathione, or 0.25× SD – consisting of 25 volumes of regular SD and 75 volumes of water containing 2% glucose). Typically, we added ~10 ml of fresh media. Each new culture contained the cells from ~1 ml of the initial culture (initial density ~500,000 cells per ml). We incubated the new cultures at the desired temperature for 14 days in glass tubes in a rotator set to 40 rpm. We performed RNA extractions using the RiboPure Yeast RNA extraction kit (Thermo Fischer Scientific, cat. No. AM1926) following the kit instructions. We also performed the DNAse treatment after RNA extraction and stored the isolated RNA in elution buffer at −80 °C before further processing.

## Measuring genome-wide transcription rate with 4tU labeled RNA (see Supplementary Text for details)

In short, we prepared large cultures of our wild-type yeast similarly to the description in the Methods section titled, "growth experiments", and added 4-thiouracil (4tU) to the growth media at a final 5 mM concentration[42]. Samples were subsequently collected after desired amounts of time at each temperature. We discarded the supernatant and re-suspended the pellet in 1 ml of RNAlater (Cat. No. AM7021, Thermo Fischer Scientific). As a spike-in of 4tU labeled RNA we used a fixed amount of cells from *Schizosaccharomyces pombe* (YFS110) analogously to previous work[42]. We then spun down our samples in a pre-cooled centrifuge, removed the RNAlater, and proceeded with RNA extraction as described in the Methods section titled, "preparing cells and RNA extraction". After RNA extraction, we proceeded with biotinylation and purification of the 4tU labeled RNA following existing protocols with minor modifications[42]. After sequencing, we processed all sequencing data with the Salmon tool (v1.5.1) to quantify relative transcript abundance[43]. Finally, we converted the transcript levels for *S. cerevisiae* to gene expression levels (Transcripts Per Million, TPM), merged all samples using the package tximport (v3.16) from Bioconductor[44] and converted our 4tU time-lapses to "# of RNA per cell" (more details in Supplementary Figs. 34–36). We also used R Studio 3.5.1 to analyze these data.

## Mathematical model

Derivations of equations, a detailed description of the mathematical model, and the parameter values used for simulations are in the Supplementary Notes.

## Codes' functionality

We provide two scripts, written in MATLAB (R2020). Running these two scripts produces Fig. 6c, d as outputs. "code_fig6c.m" requires no input or data. It produces Fig. 6c. "code_fig6d.m" requires no input or data. It produces Fig. 6d. Only a commercial license for MATLAB is required to run these MATLAB codes (license for version R2015 or later). No other licenses or software are required to run our codes. After installing MATLAB, one downloads our codes on Github (see the "Code availability" statement) and hit "run" in the MATLAB environment to execute our codes. Together, these codes implement stochastic simulations of single cells at frigid temperatures: how each cell randomly replicates, dies, or remains alive without replicating at each time step. Please see Supplementary Data 1 for the pseudocodes that describe our codes in detail.

## Statistics and reproducibility

No statistical method was used to predetermine sample size. The experiments were not randomized. No data were excluded from the analyses. The investigators were not blinded to allocation during experiments and outcome assessment. In general, sample sizes were $n = 3$ biologically independent replicates, or the nearest integer n such that the number of conditions or biological replicates fills up the available equipment (e.g., six conditions with each condition having four biological replicates for an experiment in which we used a 24-well plate). For microscopy, the number of analyzed cells were limited by the number of cells that could be in a field of view and remain unobstructed throughout the duration of the resulting, time-lapse movie.

## Reporting summary

Further information on research design is available in the Nature Portfolio Reporting Summary linked to this article.

## Data availability

Source data are provided with this paper. Data generated in this study are available in the Source Data file and at: https://github.com/youklab/LamanTrip-coldTemp-2022. The RNA-Seq data generated in this study are available at NCBI GEO database under accession code GSE211918. The Gene Ontology data for *Saccharomyces cerevisiae* used in this study are available at the YeastPathways data base https://pathway.yeastgenome.org/YEAST/NEW-IMAGE?object=Gene-Ontology-Terms. Source data are provided with this paper.

## Code availability

All scripts used for simulations in this work are available at GitHub: https://github.com/youklab/LamanTrip-coldTemp-2022/tree/main/Codes.

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

## Acknowledgements

We thank Jennifer Benanti, Amir Mitchell, Job Dekker, Marian Walhout, and Michael Lee for insightful comments and suggestions. We thank Nicholas Rhind for providing the *S. pombe* strain that we used as a control for our 4tU-Seq data. We thank Ezequiel Calvo and Athma Pai for suggesting the 4tU-Seq experiment. H.Y. was supported by the European Research Council (ERC) Starting Grant (MultiCellSysBio, #677972 to H.Y.), Netherlands Organisation for Scientific Research (NWO) Vidi Award (#680-47-544 to H.Y.), CIFAR Azrieli Global Scholars Program (H.Y.), and EMBO Young Investigator Award (H.Y.).

## Author contributions

D.S.L.T. and H.Y. conceived the project. D.S.L.T. designed and performed all experiments, analyzed data, and built the mathematical model. T.M. helped with microscope imaging. D.S.L.T. and H.Y. examined all data and discussed data analyses. H.Y. supervised the research. D.S.L.T. and H.Y. wrote the manuscript.

## Competing interests

The authors declare no competing interests.
