## [Peer Review File · Nature Communications]

Slowest possible replicative life at frigid temperatures for yeastREVIEWER COMMENTS

Reviewer #1 (Remarks to the Author):

This paper is a tour de force that interrogates under what conditions yeast can grow at low temperatures and the limits of their ability to grow. Integrating both theory and experiment, the paper also elucidates the conceptual and molecular mechanisms underlying low temperature growth or death. Specifically, like at high temperatures, yeast secrete glutathione at low temperatures. This helps drive cell replication in higher starting density populations through reducing reactive oxygen species as reactive oxygen species inhibit the replicative phase of the cell cycle. Without this reduction in reactive oxygen species cells slowly grow but do not divide and eventually burst.

I have no major concerns about this manuscript. It is highly polished, easy to read, and follows up on each claim in depth. I recommend publication and have two minor comments the authors may wish to address:

(1) Supplemental Figure S21 is hard to understand. Panel a top appears to have a cell that dies but is not marked like in panel b. In panel c, the relationship between the different duplicating populations (purple) and the populations of dying cells indicated by the arrows is unclear.

(2) Supplemental Figure S26 Could benefit from some sort of visual indication of how b is a subset of a beyond the grey shading, or perhaps the grey shading might be removed. b appears to be only the cells in G1, which would not be easy to visually mark on panel a, but the way the shading is currently done suggests it is zooming in on part of the panel.

Reviewer #2 (Remarks to the Author):

In this study, the authors examined the growth characteristics of yeast cells under frigid conditions. They made several major contributions.

1. They established the “hard” limit of the ability of cells to survive under low temperatures.
2. They demonstrated that under such conditions, survival is density-dependent. That is, the population growth exhibits Allee-effect, where a population can only survive at a sufficiently high initial density.

3. They demonstrated that the collective survival results from the production of an antioxidant (glutathione), which enables the removal of ROS, accumulated due to cells' response to cold nutrients.
4. They further demonstrated that ROS slows down growth primarily by prolonging the G1 phase of the yeast cell cycle.

Finally, they established a coarse-grained model to relate cell doubling time and the temperature, which is in part based on first-principle-based reasoning and in part constrained the experimental measurements.

This study is absolutely a tutor-de-force. It is incredibly comprehensive. It starts with a very basic question, where only vague or somewhat contradicting answers are available in the literature, in part due to a lack of well-controlled and thorough analysis. Though the question is basic, getting to the bottom of it is far from trivial. The authors made meticulous efforts to unravel this question layer-by-layer. It's scientific investigation at its best. Despite the massive amount of data and the depth of the analysis, the paper is a joy to read. The writing is crystal clear and the figures are accurate, informative, and pleasing to look at. The central conclusions of the work are robustly established.

Also, I really appreciate the quantitative reasoning that the authors used in establishing a coarse-grained model for explaining the data. As a first step, the authors made a number of simplifying assumptions: e.g., that protein synthesis rates and cell division rates follow Arrhenius law, and how the extra time depends on ROS removal (as well as the exact form of that dependence). These assumptions to me are totally appropriate; likely, the data and the model will stimulate future research (experimental and modeling) in further dissecting the mechanism underlying yeast growth under frigid conditions.

I learned a lot from reading the paper, not only in terms of the central biological insights but also in terms of the experimental/computational approaches the authors took. This study is a textbook example of how to properly conduct rigorous quantitative biology research. I believe this study will have a significant and lasting impact.

I recommend its publication in the journal.

Reviewer #3 (Remarks to the Author):

This is an interesting manuscript, which describes a comprehensive study that aims to understand how yeast cells survive and divide at low temperatures. This is a follow-up study to one that was published 2 years ago in Nat Microb, which characterized the growth of yeast at high temperatures.

Interestingly, similar to the previous study, also cold conditions lead to increased intracellular oxidation and an expression of glutathione in the medium, which is crucial for the survival and division of the yeast cells at extreme temperatures. Also, here, initial cell density defines if the culture will cultivate or not.

Thus, the authors present a very similar mechanism of cell survival under heat and cold conditions.

Moreover, the authors utilize slow growth to correlate between cell cycle progression and redox status of cells, emphasizing that accumulation of oxidants is correlated with prolonged G1 phase and most probably inhibits S-G2 phase.

The paper is very well written, easy to follow, and provides an extensive set of supplementary figures to explain the data and the phenotype.

To be useful for the community, the authors should address the following concerns.

Major comments:

1. Due to high mechanistic similarities between the proposed mechanism for cell survival during cold and heat conditions, the differences and similarities should be addressed. As well a comparison with normal conditions is missing (especially in the cell cycle analysis and ROS, GSH quantification). It is not clear if export or import or both of GSH is important for the proposed phenomenon. The authors used delta GSH strain to examine the effect of the GSH synthesis but it is not clear if the secreted GSH is the result or cause of the cell survival. In the high temperature, the authors showed that strains defected in the GSH export cannot divide while import of GSH did not have any effect. What about the cold conditions?

Seems like this is a general stress response.

2. Almost all experiments show an extremely high cell-to-cell diversity, does it mean that there are different mechanisms for coping with the cold conditions or/and different generations of cells that were analyzed. The doubling time varies from 2-17 days (Fig 1c), which is almost one order of magnitude! Importantly, this diversity is found in both low and high-density populations, even though doubling time is 3-4 fold different. The same is true for ROS accumulation (Fig 2).

Moreover, the cell cycle progression is much faster than the doubling time. It is very confusing. If indeed the cell cycle is around 2-3 days then the doubling time is measured for either new daughter cells (which

are usually reduced) with old cells (which are usually oxidized). Labeling budding scars might help to define the heterogeneity and the related oxidation. As was shown by many groups, replicative and chronological aging correlates with increased oxidation. Thus, this might be the phenomenon that was monitored here, and it is not related to the low-temperature response per se.

3. It is not clear if the glutathione that was measured here is reduced or oxidized or both. What is the GSSG/GSH ratio? Moreover, the cell images of mitochondrial and cytosolic ROS are very strange. Mitochondrial ROS looks cytosolic, and at a very high level. The method part is not sufficient to understand the quality of measurement. The images fit dead cells and not living cells. Controls of these images should be provided (e.g., living cells treated with peroxide, DTT). How do the same images look for cells grown in normal temperatures? The authors claim many times that ROS is the main reason for growth inhibition. It can be a secondary effect. A decrease in G1 can be a result of many other factors altering redox homeostasis and cellular oxidation.

4. Fig 2 Nutrients, p.7. I found this part quite speculative. It is quite obvious that adding antioxidants improves the growth rate, not only at 5 degrees. It was shown in many biological models, aging phases and conditions. Manipulation of the growth medium will definitely alter the metabolism, respiration, and oxidation rate. Reductive conditions increase doubling time in normal conditions as well. Most probably, these conditions alter the diauxic shift of the yeast cells. Thus, I find the statement “ these results show that non-sugar nutrients are major generators of ROS in frigid temp.” is not accurate and also not solid. In addition, the phrase “ROS scavengers (superoxide) “ is very confusing. Superoxides are ROS. The method part is also not clear.

Minor comments:

1. The Introduction and abstract : there are many speculative statements and not clear phrases that do not reflect the paper.

What do “system level design principles that dictate the pace of life... “ really mean.

“we discovered “key reason” - I am still not really convinced that ROS is the key reason. It can be a result of stress conditions, failure in metabolism and many reasons. ROS was measured but it is not the key reason for survival at 5 degrees.

“we discovered that all these effects of ROS are due to one mechanism: ROS elongates the G1 ..” – again, there is more than one mechanism for sure, and maybe elongation of G1 results in increased ROS. The experiments presented here do not show direct regulation.

2. “Fig S12: “oxydation” should be oxidation.

3. “ laboratory-standard (“wt”) yeast strain – there are many “standard” strains. Should be specified.

4. Fig 3 what is the difference between “begins life” and “duplicates” – look very similar, both are in the budding phase.

List of major changes made:

- **Added 14 new supplementary figures (total of 47 Supplementary Figs).**
Additional data to satisfy Reviewer 3's requests. No conclusions have changed. All new data are in the supplement because they are tangential to the main storyline. Some new data are solely in this letter (see response to Reviewer 3). Renumerated the Supplementary Figures to account for these additions. *The new supplementary figures are: 11, 15, 20, 28, 29, 32, 33, 37-40, and 44-46.*
- **Added "Supplementary Discussion" on Pgs. 84-85 of Supplementary Information.**
Here, we compare how yeast behaves at low temperatures (below 8 °C) to how they behave at high temperatures (above 37 °C). These are based on both the new data from the revision and the data that we submitted before the revision. We added this new section as a response to Reviewer 3.
- **Shortened the main text to ~6000 words to meet the journal's word-count limit.**
We shortened the text primarily by shortening the Introduction and Discussion sections. We hardly changed any of the Results section. This shortening also prevents us from expanding the scope of our work further than this revision has already done (Reviewer 3's major comment 1 further expands the manuscript's scope which we have partly done during this revision).
- **Shortened the abstract to meet the word-count limit.**
- **Shortened the title to meet the word-count limit.**
- **Minor, stylistic changes in accordance with the journal's guidelines.**

Response to Reviewer 1

Thank you very much for the positive, overall assessment of our work. Below, *the reviewer's comments are in italics* and are followed by our response. **Highlighted in yellow are the changes that we made in our manuscript to address the reviewer's comments.**

"This paper is a tour de force that interrogates under what conditions yeast can grow at low temperatures and the limits of their ability to grow. Integrating both theory and experiment, the paper also elucidates the conceptual and molecular mechanisms underlying low temperature growth or death. Specifically, like at high temperatures, yeast secrete glutathione at low temperatures. This helps drive cell replication in higher starting density populations through reducing reactive oxygen species as reactive oxygen species inhibit the replicative phase of the cell cycle. Without this reduction in reactive oxygen species cells slowly grow but do not divide and eventually burst.

I have no major concerns about this manuscript. It is highly polished, easy to read, and follows up on each claim in depth."

We thank the reviewer for this positive, overall assessment of our work. We are very grateful for this positive assessment.

"I recommend publication and have two minor comments the authors may wish to address:

(1) Supplemental Figure S21 is hard to understand. Panel a top appears to have a cell that dies but is not marked like in panel b. In panel c, the relationship between the different duplicating populations (purple) and the populations of dying cells indicated by the arrows is unclear."

We added clarifying remarks in the caption for Fig. S21 (now Supplementary Fig. 24). Specifically,

- Supplementary Fig. 24a: cells - represented by the green or red curve - don't die here. The green and red curves in the top graph terminate after ~7 days because, due to nearby cells dividing, the cells represented by these curves become occluded by newborn cells that come into the same field of view.
- Supplementary Fig. 24c: this figure shows that the non-replicating cells - cells that either die (orange bar) or increase in size without duplicating (grey bar) - can be grouped into

either being in G1 (top arrow) or as currently duplicating (bottom arrow), according to the nuclear Whi5 measured in the replicating cells. Note that we measured the nuclear Whi5 in both the G1 and S/M phase of each cell. The two purple bars are not from two different populations. They are made from the same population of duplicating cells, with each cell belonging to one of the two - but not both - bars. The left purple bar shows the average nuclear whi5 for cells that eventually duplicate when these cells are in G1. The right purple bar shows the average nuclear whi5 for cells that eventually duplicate when these cells are in the S/M phase. **To avoid confusion, we removed the two arrows in this figure and modified the caption.**

"(2) Supplemental Figure S26 Could benefit from some sort of visual indication of how b is a subset of a beyond the grey shading, or perhaps the grey shading might be removed. b appears to be only the cells in G1, which would not be easy to visually mark on panel a, but the way the shading is currently done suggests it is zooming in on part of the panel."

We followed the reviewer's suggestion by removing the grey shading and the grey box that encloses the graphs in Fig. S26 (now Supplementary Fig. 31). We changed the label, "grows", to "grows without duplicating" in both (a) and (b). The blue (green) bars in (b) show the percentage of cells from the blue (green) populations in (a) that remained in G1. For example, if a cell dies in a population without the added GSH (blue bar in (a)), then it was ~80% likely to be in G1 phase (blue bar in 'dies' in (b)).

Thank you so much for taking the time to evaluate our manuscript. Your comments have improved our manuscript by helping us clarify the main concepts.

Response to Reviewer 2

Thank you very much for the positive, overall assessment of our work. Below, *the reviewer's comments are in italics* and are followed by our response. **Highlighted in yellow are the changes that we made in our manuscript to address the reviewer's comments.**

"In this study, the authors examined the growth characteristics of yeast cells under frigid conditions. They made several major contributions.

- 1. They established the "hard" limit of the ability of cells to survive under low temperatures.*
- 2. They demonstrated that under such conditions, survival is density-dependent. That is, the population growth exhibits Allee-effect, where a population can only survive at a sufficiently high initial density.*
- 3. They demonstrated that the collective survival results from the production of an antioxidant (glutathione), which enables the removal of ROS, accumulated due to cells' response to cold nutrients.*
- 4. They further demonstrated that ROS slows down growth primarily by prolonging the G1 phase of the yeast cell cycle.*

Finally, they established a coarse-grained model to relate cell doubling time and the temperature, which is in part based on first-principle-based reasoning and in part constrained the experimental measurements.

This study is absolutely a tutor-de-force. It is incredibly comprehensive. It starts with a very basic question, where only vague or somewhat contradicting answers are available in the literature, in part due to a lack of well-controlled and thorough analysis. Though the question is basic, getting to the bottom of it is far from trivial. The authors made meticulous efforts to unravel this question layer-by-layer. It's scientific investigation at its best. Despite the massive amount of data and the depth of the analysis, the paper is a joy to read. The writing is crystal clear and the figures are accurate, informative, and pleasing to look at. The central conclusions of the work are robustly established.

Also, I really appreciate the quantitative reasoning that the authors used in establishing a coarse-grained model for explaining the data. As a first step, the authors made a number of simplifying assumptions: e.g., that protein synthesis rates and cell division rates follow Arrhenius law, and how the extra time depends on ROS removal (as well as the exact form of that dependence). These assumptions to me are totally appropriate; likely, the data and the model will stimulate future research (experimental and modeling) in further dissecting the mechanism underlying yeast growth under

frigid conditions.

I learned a lot from reading the paper, not only in terms of the central biological insights but also in terms of the experimental/computational approaches the authors took. This study is a textbook example of how to properly conduct rigorous quantitative biology research. I believe this study will have a significant and lasting impact.

I recommend its publication in the journal."

We thank the reviewer for this positive, overall assessment of our work. We are very grateful for these detailed comments.

Thank you so much for taking the time to evaluate our manuscript and for your detailed and positive assessment of both the mathematical reasoning and experimental results in our work.

Responses to Reviewer #3:

Thank you for the positive, overall summary of our work that begins your report. Below, *the reviewer's comments are in italics* and are followed by our response. **Highlighted in yellow are the changes that we made in our manuscript to address the reviewer's comments.**

We believe that many of the reviewer's comments stem from unfortunately misunderstanding our mathematical or logical reasonings.

As we highlight below, we believe that three of the four major comments (major comments 2-4) stem from one of two misunderstandings:

1. When we say that "X primarily causes Y" or that "X primarily determines Y", we're **not** saying that "X is the only cause for Y" or that "X is the only determinant of Y".
2. Single-cell-level values don't reflect population-level average: cell-doubling time and ROS level can markedly differ among cells and do not reflect the population-level average.

To prevent such misunderstandings, we added a "Supplementary Discussion" section on Pgs. 84-85 of Supplementary Information. By paraphrasing our responses below, this section clarifies the above issues and responds to reviewer's major comment 1.

"This is an interesting manuscript, which describes a comprehensive study that aims to understand how yeast cells survive and divide at low temperatures. This is a follow-up study to one that was published 2 years ago in Nat Microb, which characterized the growth of yeast at high temperatures.

Interestingly, similar to the previous study, also cold conditions lead to increased intracellular oxidation and an expression of glutathione in the medium, which is crucial for the survival and division of the yeast cells at extreme temperatures. Also, here, initial cell density defines if the culture will cultivate or not.

Thus, the authors present a very similar mechanism of cell survival under heat and cold conditions.

Moreover, the authors utilize slow growth to correlate between cell cycle progression and redox status of cells, emphasizing that accumulation of oxidants is correlated with prolonged G1 phase and most probably inhibits S-G2 phase.

The paper is very well written, easy to follow, and provides an extensive set of supplementary figures to explain the data and the phenotype."

Thank you for this positive, overall summary.

We appreciate the positive overall summary. But we respectively disagree with the reviewer's claim that our work is a follow-up of our earlier paper on high temperatures (temperatures higher than 36.5 °C). Therefore, we respectfully disagree that we should be re-doing experiments from our earlier paper but now at frigid temperatures or that we should repeat experiments from our current manuscript but now at high temperatures as the reviewer suggests in their subsequent comments. The reasons for our disagreement are as follows:

First, we are not sure what the reviewer means by "the phenotype" here. Our systems-biology work describes many interconnected phenomena that include: glutathione secretion (Fig. 2); effects of knocking out enzymes that reduce ROS without glutathione (Fig. 4); Whi5 deletion enabling cells to duplicate even with abundant ROS (Figs. 4 & 5); quantitative relationships between a cell's ROS level and its ability to grow, its ability to duplicate, and its likelihood of dying (Fig. 3); and how the global transcriptional rate and protein-synthesis rate - along with ROS-reducing ability that has nothing to do with glutathione - together determine a cell's doubling time and "speed limits" at every frigid temperature (Figs. 5 & 6). Most of these phenomena weren't examined in the high-temperature work. Our work doesn't solely focus on our discovery of yeast secreting glutathione at temperatures below 8 °C. The glutathione secretion is mentioned only in Fig. 2. After this figure, we move onto establishing the other phenomena mentioned above. Many of these phenomena depend on ROS, independently of glutathione's involvement. Thinking of our manuscript to be primarily about yeast secreting glutathione at frigid temperatures would be misconstruing our work as a follow-up of the high-temperature work. Our work is also not a comparative study that is meant to accompany the high-temperature work.

Secondly, we disagree that our work is a follow-up of our high-temperature paper for the following, additional reasons:

- It does not logically follow that yeast must secrete glutathione at temperatures below 8 °C just because they do so at temperatures above 36.5 °C (especially because our earlier paper showed that yeast does not secrete glutathione at temperatures between 30 °C and 36.5 °C). In the current study, we discovered that yeast secretes glutathione at frigid temperatures ($T < 8$ °C) by making a lucky guess. No logical deductions from the high-temperature work would have led to this discovery.
- Glutathione secretion - the sole focus of the high-temperature work - is only a minor portion of our manuscript. Instead, our manuscript primarily focuses on ROS - which doesn't require glutathione for us to think about - and gene-expression speed at low temperatures. None of the aspects shown in Figs. 3-6 are in the high-temperature paper. We also used techniques that weren't used in the high-temperature work (e.g., single-cell microscopy, 4-tU RNA-Seq, new mutants that cannot reduce ROS, cell-cycle mutants, etc.).
- None of the results in our manuscript rely on or require any knowledge of the high-temperature work. Two different studies, in two different environmental conditions, can find some of the same molecules and pathways as being important. But that doesn't mean that one is a follow-up of the other study.

Our goal was, at frigid temperatures, to discover how slowly yeast can complete its cell cycle and to reveal any fundamental barriers to slowing down the replicative dynamics of yeast. It might be fruitful for another paper - a follow-up paper - to compare the molecular mechanisms at the high-temperature regime with those of the low-temperature regime. But doing so is not required for any of our conclusions and the main storyline. Moreover, already at 47 Supplementary Figures, doing so would push our work enormously outside the reasonable scope of a single paper. Please note that such comparative study wouldn't even be possible without the discoveries that our current manuscript reports.

"To be useful for the community, the authors should address the following concerns.

Major comments:

1. Due to high mechanistic similarities between the proposed mechanism for cell survival during cold and heat conditions, the differences and similarities should be addressed. "

For reasons that we mentioned above, our work is neither a follow-up of nor a comparative study for the high-temperature paper. Thus, we respectfully disagree that we should repeat the experiments that were done in the high-temperature paper but now at frigid temperatures or that we should repeat the experiments that are in our manuscript but now at high temperatures. Doing so is also unnecessary given that the reviewer states that our work is already "*backed up by an extensive set of supplementary figures*" and given that Reviewers 1 & 2 both state that our work is already a "*tour-de-force*". Please also note that as it is, we had to cut the number of words down to 6000 words - the maximum number of words allowed by the journal.

Although we disagree with expanding the scope of our work, we still performed additional experiments and added more data in response to Reviewer 3 over the last two months. We can use the additional data (some of the 14 new Supp. Figs.) and the data from the original submission to distinguish the yeast's behavior at frigid temperatures from yeast's behavior at high temperatures. Furthermore, on pgs. 84-85 of Supplementary Information, we added a "Supplementary Discussion" section to discuss the similarities and differences between the frigid- and high-temperature phenomena (paraphrasing of our responses below).

Differences between the frigid- and high-temperature phenomena: we uncovered myriad phenomena at frigid temperatures that one couldn't logically deduce from reading our high-temperature paper. We discovered some of these by using mutants that weren't used in the high-temperature paper. With these results, we can already answer the reviewer's comment above by distinguishing frigid- and high-temperature phenomena. We added a "Supplementary Discussion"

section to explain these differences (rephrasing of the items mentioned below). The main differences are:

1. **GSH is sufficient but not necessary** to induce population growth at frigid temperatures whereas GSH alone is both sufficient and necessary to induce population growth at high temperatures. **We establish this with data from additional experiments (Supplementary Fig. 11) in which we used a masking agent to block extracellular GSH.** This observation is further supported by the fact that removing non-sugar nutrients from the growth medium also induces population growths at frigid temperatures (Fig. 2g) but not at high temperatures. Thus, it is not necessary to investigate GSH-import mutants because extracellular GSH is not necessary - it is only sufficient - for the cooperative growth of yeast at frigid temperatures.
2. Additionally, at high temperatures, the doubling time of a low-density population that grows with supplemented GSH is very similar to that of a high-density population that grows without a supplemented GSH. In contrast, at frigid temperatures, adding GSH not only enables low-density populations to grow, increasing its concentration can shorten the population's doubling time by several folds (Fig. 2e).
3. Furthermore, at high temperatures, GSH's growth-promoting actions are solely extracellular since GSH export is essential for populations to grow while GSH import is not necessary for population growths. In contrast, at frigid temperatures, GSH seems to have at least some special intracellular roles for growth given that mutants that lack certain ROS-reducing enzymes - enzymes that use GSH as a co-factor - have severely limited growth (Fig. 4e).
4. Finally, at high temperatures, high-density populations accumulate extracellular GSH up to the amounts that we must add to the medium of a low-density population to induce growth of that low-density population. In contrast, at frigid temperatures, high-density populations cannot accumulate extracellular GSH to the amounts that we must add to the low-density population to induce the growth of that low-density population. This is because, besides GSH, other factors such as non-sugar nutrients (Figs. 2f-2i) and intracellular processes play a role in determining whether a population grows at frigid temperatures.

In addition to these differences, to further satisfy the reviewer's request for additional work without substantially expanding the scope of our work, **we added data from additional experiments that further show how yeast behaves differently at frigid temperatures compared to at high temperatures.** Among these, the most relevant are:

1. **Supplementary Fig. 29:** knocking out some of the major heat-shock proteins - known to be important for combatting heat-induced damages at high temperatures - does not severely affect population growth in frigid temperatures.
2. **Supplementary Fig. 33:** at 1 °C, adding a saturating level (250 μM) of GSH cannot induce population growths and does not decrease intracellular ROS levels whereas it can achieve both at 5 °C. This result, like the others in our manuscript, again emphasizes why our work isn't about GSH but is about ROS and the ways that cells can duplicate with abundant ROS at frigid temperatures (by lacking Whi5 as shown

in Figs. 4-5). This is different from the high-temperature work in which we showed that adding sufficiently high amounts of GSH is sufficient to cause population growths and extends the range of habitable temperatures to higher values.

"As well a comparison with normal conditions is missing (especially in the cell cycle analysis and ROS, GSH quantification)."

We already provided comparisons to "normal conditions" in all our experiments. The "normal condition" - a "normal cell" to be correct - is a cell that can survive and divide **at the same frigid temperature** as the cells that die and/or cannot divide. In other words, a "normal cell" that we need to compare to is a cell that acts "normally" (i.e., is healthy) at the same frigid temperature. A "normal" cell is thus a cell that has relatively low levels of ROS at frigid temperatures (at or below 5 °C). A "normal" cell is not a cell at 30 °C because a cell at 30 °C might still have a relatively high level of ROS. To show ROS' effect on a cell's ability to grow, duplicate, or survive, we must fix all other variables, including temperature (e.g., fixed at 5 °C), while only varying the one variable of interest - the intracellular concentration of ROS - to see how ROS affects cells.

Specifically, our cell-cycle analysis compares cells with high and low levels of ROS – with and without added GSH – and compares cells that can duplicate with cells that are unable to duplicate. For quantifying the extracellular concentration of GSH, we again compared growing and non-growing populations (Figs. 2a-2b) and performed the comparisons across temperatures which included 30 °C (Supplementary Figs. 8 & 33).

For ROS quantification, we compared non-growing populations with growing populations, with and without added GSH. Especially for quantifying intracellular ROS levels, a comparison with cells at 30 °C would not make sense for our study because the time required for the ROS dye to stain alive cells (20-30 mins) is too long compared to the cells' doubling time at 30 °C (~90 mins). Therefore, at 30 °C, the ROS-staining wouldn't be a snapshot of the cell's state but rather some convoluted report of a state that results from integrating over the cell cycle. This is in stark contrast to the staining that we performed at frigid temperatures in which the doubling time was anywhere from days to weeks (thus the staining gave a true snapshot of a cell at a particular location in a cell cycle). Moreover, cellular processes occur faster at 30 °C compared to frigid temperatures, as we have quantified. Hence, cells at 30 °C may accumulate more ROS than cells at frigid temperatures during the same time-window due to, say, cells at 30 °C having higher metabolic activities. Finally, the oxidation rate of the dye is probably temperature dependent, such that the different amounts of fluorescence may be measured for the same amount of ROS at different temperatures. **Despite these issues making the requested experiment technically problematic, not rigorous, and disconnected from our study, we still performed new experiments and quantified the ROS levels in cells at 30 °C as explained in one of our subsequent responses (see Fig. R4).**

"It is not clear if export or import or both of GSH is important for the proposed phenomenon. The authors used delta GSH strain to examine the effect of the GSH synthesis but it is not clear if the secreted GSH is the result or cause of the cell survival. In the high temperature, the authors showed that strains defected in the GSH export cannot divide while import of GSH did not have any effect. What about the cold conditions?"

Respectfully, we point out that we didn't use only the delta-GSH strain to examine how yeast is affected by glutathione and, more importantly, by ROS (regardless of glutathione). We used multiple mutants - many of which we didn't use in the high-temperature paper - to examine the effects of ROS and GSH. Below is a summary of these results that address the reviewer's question:

1. Extracellular GSH causes cells to survive because adding GSH to the growth media rescues a low-density population from becoming extinct by causing that population to grow until the culture saturates at 5 °C (see Figs. 2c & 2e).
2. GSH secretion is also a result of cells surviving because dead cells do not synthesize and do not secrete GSH at frigid temperatures.

Points 1 & 2 answer the reviewer's question: extracellular (secreted) GSH causes yeast to survive and, at the same time, results from yeast surviving. Furthermore,

3. GSH is sufficient but not necessary at frigid temperatures whereas it is both sufficient and necessary at high temperatures. Supplementary Fig. 11 establishes this by showing that removing the secreted GSH with a masking agent ("undoing glutathione export") still permits population growth (i.e., glutathione export is unnecessary) (also see our Point 1 at the top of Pg. 9 of this letter). **This is the most relevant difference regarding GSH between the two temperature-regimes. Page 84 of the Supplementary Information now includes a "Supplementary Discussion" section to mention this.**
4. As GSH is not necessary for population growths at frigid temperatures, there are other mechanisms that, by affecting cell's ROS level, induce population growths at frigid temperatures. For example, we have shown that removing non-sugar nutrients also induces population growth (Fig. 2g). Also, we established that adding scavengers of extracellular ROS can also accelerate population growth (Supplementary Fig. 13). Import of GSH is thus not relevant: we have other means of enabling cell duplications at frigid temperatures (i.e., glutathione import is irrelevant).

Because of points 3 & 4, focusing on import and export of GSH would have been inappropriate for our study: doing so would not have led us to the heart of the matter which is ROS and its impact on cells at frigid temperatures. GSH is just one dial that tunes cells' ROS levels. That's why we focused on ROS and its impact on cells (Fig. 3), rather than on GSH, and

built the mutants that affect intracellular ROS. For example, we knocked out ROS-reducing enzymes, and we knocked out *WHI5* which controls the ROS' gating of G1-to-S transition.

"Seems like this is a general stress response."

Whether GSH secretion is part of some "general stress response" is not the focus of our work and has no bearing on the logical flow and conclusions of our work. Moreover, there is neither a rigorous definition of nor a universal consensus on what a "general stress response" truly is for yeast. There are various views on this topic. If the reviewer's claim is that GSH secretion is part of yeast's "general stress response" because we discovered that the secretion occurs for temperatures below 8 °C and for temperatures above 36.5 °C, then we take no issues with that. But we do not make such a claim anywhere because doing so isn't rigorous, is not necessary to move our study forward, and is thus not in the character of our rigorous, quantitative study.

"2. Almost all experiments show an extremely high cell-to-cell diversity, does it mean that there are different mechanisms for coping with the cold conditions or/and different generations of cells that were analyzed. The doubling time varies from 2-17 days (Fig 1c), which is almost one order of magnitude! Importantly, this diversity is found in both low and high-density populations, even though doubling time is 3-4 fold different."

There isn't any contradiction here. The key to resolving this unfortunate misunderstanding is that a population-level measurement doesn't necessarily reveal how individual cells in that population behave (e.g., some cells in a population can divide - perhaps taking 2 days or 17 days to divide - but that population will still become extinct if more cells die per day than divide per day (i.e., if death rate is higher than birth rate)). Crucially, the percentage of cells that divide and the individual doubling times of the dividing cells together dictate the doubling time of the population (i.e., average population-level doubling time = time taken for the number of cells in the population to double).

In more detail: fig. 1c shows the distribution of doubling times of individual cells that do divide - dying or non-dividing cells cannot be included in this distribution as they do not have a doubling time (because they don't divide). In contrast, the population growth curves in Fig. 2 are the result of both the percentage of cells that divide per unit time and the doubling times of the dividing cells. To illustrate, the manuscript contains a Supplementary Note describing how the density effect emerges. In short,

- **For a low population-density (Fig. 1c - bottom):** 17% of the cells in the population duplicates with an average doubling time of 7.1 days. The 17% of cells can therefore duplicate 3 times or less during our 20-day time-lapse imaging (Fig. 1c). The duplicating

cells thus expand the population by at most 52%, by creating newborn cells. During the same time-window, 60% of the cells in the population dies. The number of dying cells exceeds the number of newborn cells in the same time-window. This leads to the low-density population eventually becoming extinct.

- **For a high population-density (Fig. 1c - top):** 29% of the cells in the population duplicates with an average doubling time of 6.5 days. The 29% of cells can therefore duplicate at least 3 times during our 20-day time-lapse imaging (Fig. 1c), thereby expanding the population by 88% with newborn cells. During the same 20 days, 36% of the cells in the population from the start of the time-lapse imaging died. Thus, the number of cells that die is less than the number of cells created (newborns) in the same time-window. This leads to the high-density population growing.

Thus, the populations can differ by several folds in their doubling times despite individual cells of both populations having identical doubling-time distributions (Fig. 1c).

Continued from above:

".....does it mean that there are different mechanisms for coping with the cold conditions or/and different generations of cells that were analyzed"

No, it does not mean that. Please see our previous response. Just because cells in the same population take different durations to divide does not mean that all or any of those cells have different mechanisms to divide and/or cope at frigid temperatures (or at any temperature). In fact, in almost any microbial or mammalian cell cultures, it would be bizarre not to see cell-to-cell variability in doubling times even at the optimal growth temperature (e.g., 100 genetically identical cells in the same population will not all take exactly 2 hours to divide). Our data and mathematical model establish that ROS is the main factor that we need to know to determine the cell's doubling time, survival probability, duplication probability, duration of each cell-cycle phase, and any other aspects of that cell that we investigated.

Moreover, by growing a high-density population to saturation and then diluting that population into a fresh medium so that it begins as a low-density population, we established that this low-density population does not grow at all. **This shows that our experiments cannot be explained by the population generating subpopulations of mutants or cold-tolerant resistors.**

"The same is true for ROS accumulation (Fig 2)."

Please see our previous responses. There isn't any inconsistency here.

"Moreover, the cell cycle progression is much faster than the doubling time. It is very confusing."

On the contrary, the cell-cycle progression (Fig. 4d) is not much faster than the shortest doubling times realizable by individual cells that divide (e.g. the 'peak' of the distributions in Fig. 1c). For example, with the added GSH, the replicative phase of the cell cycle exceeds two days and, together with the duration of G1 (2.8 days total), matches the measured single-cell doubling times (2.9 days, Supplementary Fig. 12). For other cells, the replicative phase of the cell cycle is relatively fast compared to the doubling time because the replicative phase does not include the G1-phase duration. It is the G1-phase duration that varies widely among cells (Fig. 4c).

Additionally, we point out that the cell cycle progression and single-cell doubling times are very different measurements - one being population-level average and the other being single-cell-level measurements. Moreover, the two measurements were taken with different strains with different genetic backgrounds. Fig. 4d describes the average duration of each cell-cycle phase in a population (averaged over many cells). We measured these average durations with fluorescent markers. These durations can be added up to give the average duration for cell-cycle progression as suggested by the reviewer. But this sum - being merely an average over all cells in a population - would not reflect the individual doubling time of any single cell (e.g., an average family has 1.9 children, but no actual family has 1.9 children). In contrast, Fig. 1c and other single-cell level measurements of each cell's doubling time describe the doubling time of single cells being the duration between, say, bud formations.

"If indeed the cell cycle is around 2-3 days then the doubling time is measured for either new daughter cells (which are usually reduced) with old cells (which are usually oxidized). Labeling budding scars might help to define the heterogeneity and the related oxidation. "

Respectfully, we point out that there is a misunderstanding of the mathematical reasoning here which may be due to the same misunderstanding as in the reviewer's previous comment. We split the cell cycle into two parts, the replicative phase (2-3 days) and the G1 (growth) phase. The doubling time is given by the sum of these two parts. The G1 phase duration widely varies between cells and can thus extend the doubling time far beyond 3 days (Fig. 4c).

For example, we found that consecutive divisions in individual cells slow down at frigid temperatures (Supplementary Fig. 26), supporting the idea that an individual cell duplicates faster

when it is newly born compared to when it is older (Fig. 4d). These observations do not establish, nor do we claim, that new daughter cells have a small doubling time and that old cells have a large doubling time – only that an individual cell duplicates faster when it is newly born compared to when it is older.

In short, we agree that cells in a population are heterogenous – we did not claim otherwise anywhere in our manuscript and variability among isogenic cells is a well-established phenomenon in systems biology – and that there are differences in age, oxidative state, bud scars, etc. The exact sources of these differences within a population are not important for our study and these cell-to-cell variabilities do not impact our conclusions because we always compared different populations (e.g., low density versus high density, with added GSH versus without added GSH, etc) that were sampled from the same stock culture and therefore have the same heterogeneity of cells. Identifying all other sources of cell-to-cell variability in the doubling time is beyond the scope of our study.

"As was shown by many groups, replicative and chronological aging correlates with increased oxidation. Thus, this might be the phenomenon that was monitored here, and it is not related to the low-temperature response per se."

Our data agree with the reviewer's statement here so we're not at all disagreeing here. If we paraphrase the reviewer's statement, the reviewer is saying that "replication and chronological aging positively correlate with oxidation level in cells and are not due to some other low-temperature stress". Indeed, Fig. 3b shows, for example, that replicative ability decreases with increasing oxidation. In other words, there isn't some mysterious, general (low temperature) stress response that's preventing cell division at frigid temperatures. More generally, our work shows that ROS is indeed the main factor for aging, cell division, death, and speed of cell divisions at frigid temperatures.

In reference to one of the reviewer's other comments, this is also one of the reasons for not performing the 'normal' experiments at 30 °C and, instead, comparing different conditions at the same, low temperature (i.e., conditions resulting in a comparison between high and low oxidation states).

"3. It is not clear if the glutathione that was measured here is reduced or oxidized or both. What is the GSSG/GSH ratio?"

It is the reduced form of glutathione, GSH, that is important for reducing ROS and inducing population growth at 5 °C. That is why we write "GSH" in the main figures and say that we added "GSH" instead of writing "GSSG". As is the convention, GSH means the reduced form of glutathione.

Moreover, we measured **both** GSSG and GSH, separately from each other and their ratios in our yeast cultures:

- Supplementary Fig. 10b: see labels "Reduced (GSH)" and "Oxidized (GSSG)". This clearly shows that population grows when we give them reduced glutathione (GSH) but not when we give them the oxidized glutathione (GSSG).
- Supplementary Fig. 10c: see labels "Reduced (GSH)" and "Oxidized (GSSG)". This shows that cells extracellularly accumulate both GSH and GSSG. GSH accumulates in higher amounts and faster than GSSG. Accumulation of GSSG makes sense: GSH becomes GSSG after it reduces ROS.

"Moreover, the cell images of mitochondrial and cytosolic ROS are very strange. Mitochondrial ROS looks cytosolic, and at a very high level. The method part is not sufficient to understand the quality of measurement. The images fit dead cells and not living cells. Controls of these images should be provided (e.g., living cells treated with peroxide, DTT). How do the same images look for cells grown in normal temperatures?"

The red cells shown in Fig. 2b are NOT dead: they are alive. Cells that look like this continue to grow and can divide in our time-lapse movies. We know this because we continuously monitored these cells with time-lapse microscopy after staining the cells with the ROS dye (see Fig. R1 below). In fact, this continual monitoring is how we obtained Fig. 3b: probability of dying, growing, duplicating as a function of [ROS] in a cell).

In our manuscript, the selected pictures (red cells) in Fig. 2b are of cells with especially high [ROS] (as indicated by the red triangle connecting the leftmost picture (labeled "mitochondrial superoxide") with a red dot in the scatter plot that belongs to a high intracellular [ROS] (scatter plot in Fig. 2b). These cells were also still alive when the pictures were taken.

The alive cells indeed have mitochondrial ROS dye that nearly looks cytosolic (nearly painting the entire cell) for two reasons:

1. Our microscope, like every wide-field microscope, has a resolution limit: it cannot resolve fine structures of mitochondria with the magnification that we used.
2. Images in Fig. 2b show a projection of multiple z-slices onto a single xy-plane, thereby merging red structures that are visible in each z-slice into a single blob.

Despite the resolution-limit (diffraction limit) preventing us from seeing the finer structures of mitochondria and making us see red fluorescence bleeding out of mitochondria and smearing over the cytoplasm, the key point is that we care about quantifying the average intensity of the fluorescence dye in a cell for which we don't need to see mitochondrial structures (e.g., with a super-resolution microscope). Averaging over all pixels in a cell to get a single value - average fluorescence - doesn't require such a high-resolution and, more importantly, we do see a distribution of red-dye fluorescence with our imaging method (see Fig. 2b: scatter plot shows a wide distribution of intracellular [ROS]). If these images were just artifacts, it also doesn't support that these red-dye fluorescence positively/negatively correlate so well with the probabilities of a cell replicating and of dying (Fig. 3b).

To emphasize point #2, Fig. R2 below shows the individual z-slices of alive cells at 5 °C, stained with the mitochondrial ROS dye, which show finer, granular structures inside a cytoplasm instead

of painting the entire cytoplasm. Merging these z-slices (~30 z-slices) into one plane ("z projection") yields the types of images seen in Fig. 2b.

As a comparison, Fig. R3 below shows mitochondrial ROS staining of dead cells at 5 °C. These don't display any granular structures in individual z-slices and clearly look very different from the alive cells stained with mitochondrial ROS in Fig. R2. In these dead cells, the entire cell (including nucleus) looks completely stained and is so bright in the red channel that we can see "rays" of fluorescence diffracting out (emanating) from the cell in every z-slice.

In summary, our explanations above and Figs. R1-R3 show that there's nothing wrong with our mitochondrial ROS staining and that we stained alive cells having different levels of ROS. Moreover, note that we gave the ROS-reducing GSH to cells which directly resulted in the distribution ROS levels shifting to lower values of ROS (compare Fig. 2b - top with bottom scatter plot). **This is an experiment analogous to the one suggested by the reviewer: treating cells with GSH (an antioxidant) instead of treating them with DTT or peroxide.** The latter, inducing ROS with peroxides as suggested by the reviewer, would not be a sensible experiment for cells that already have high ROS levels. This further proves that the mitochondrial ROS dye is faithfully reporting the ROS level in each cell. Finally, we performed a similar staining with a different ROS indicator dye and obtained very comparable results (Supplementary Fig. 9 and below).

Regarding the reviewer's suggestion of doing the ROS staining at "normal temperatures" (e.g., 30 °C) as a "control": this experiment would tell us how cells' ROS levels at 30 °C compare

to those at 5 °C. But it does not logically follow that ROS levels at 30 °C would tell us whether a cell with a particular level of ROS staining at 5 °C is alive (our experiment of adding GSH to reduce ROS and then getting less ROS-dye fluorescence at 5 °C is the real experiment that's required and we've already done that). Thus, the suggested experiment is tangential to our main storyline.

Moreover, as mentioned in one of our previous responses, there are several concerns about comparing ROS staining at different temperatures:

1. At 30 °C, the time taken for the dye to stain alive cells (20-30 mins) is too long compared to the cells' doubling time (90 mins). The staining thus wouldn't be a snapshot of the cells' ROS state but rather some convoluted state that results from temporally integrating over the cell cycle at 30 °C. This situation is in stark contrast to the staining that we performed at frigid temperatures, in which the cells have doubling times ranging from days to weeks. Therefore, at frigid temperatures, we really obtain a snapshot of a cell's ROS level while the cell is at a particular location in the cell cycle.
2. Cellular processes occur more rapidly at 30 °C than at frigid temperatures, as we have quantified. Hence, cells at 30 °C may accumulate more ROS than the cells at 5 °C during the same time-window due to, say, the cells at 30 °C having higher metabolic activities. This would lead to artificially inflating the ROS-dye fluorescence level at 30 °C compared to frigid temperatures.
3. The oxidation rate of the dye likely depends on temperature, such that two cells, having the same amount of ROS but at two different temperatures, would give off two very different levels of fluorescence. This would mean that we cannot use the dye's fluorescence to reliably compare ROS level of cell at 30 °C with a ROS level of a cell at a frigid temperature.

Despite the above concerns making the requested experiment being technically problematic and irrelevant for our main conclusions, we've performed the requested experiment and, on top of that, another experiment: we stained cells at 30 °C with the mitochondrial ROS dye in one experiment and with the cellROX dye in another experiment (see Fig. R4a below). These additional experiments showed that we get the same distribution of ROS-dye fluorescence levels at 30 °C in a cell population as we do at 5 °C (with both the mitoSOX and cellROX dyes). Given our concerns listed above, we do not want to draw any conclusions from this data. Besides, any conclusions from this data also wouldn't affect any of our original conclusions.

Fig. R4: Stains with ROS dyes (cellROX on left and mitoSOX on right) at 30 °C compared to 5 °C. **a**, new data showing distribution of ROS levels in single cells at 30 °C for a healthy, growing population. **b**, data from Supplementary Fig. 9 for the distribution of ROS levels in cells at 5 °C for three different types of populations (green and blue are growing populations due to low ROS levels, red is for non-growing population). Comparing the 1st row with 2nd and 3rd rows shows that the distribution of ROS-dye fluorescence are similar between 30 °C and 5 °C for a growing population (especially compared to populations that received ample GSH (green) at 5 °C).

Finally, we point out that we compared cell sizes across temperatures that included 30 °C (Supplementary Fig. 32) and that we measured both the intracellular ROS concentration and quantified the extracellular GSH at 1 °C (Supplementary Fig. 33).

"The authors claim many times that ROS is the main reason for growth inhibition. It can be a secondary effect. A decrease in G1 can be a result of many other factors altering redox homeostasis and cellular oxidation."

There's an unfortunate misunderstanding here which underlies several comments of the reviewer. We repeat the same logical argument for each comment for completeness (but that has made this rebuttal quite long).

First, we respectfully point out that the reviewer's last sentence, "... altering redox homeostasis and cellular oxidation" is inconsistent with the reviewer's first sentence which says that ROS is a secondary effect. If redox homeostasis and cellular oxidation are messed up, then a cell's ROS level would be altered (increased or decreased). ROS level is both a reporter of and a result of redox homeostasis and cellular oxidation. Many events can indeed alter the ROS level in a cell and we do not claim otherwise anywhere in our manuscript. We do not say anywhere that only one or two events affect ROS. In our study, the intracellular ROS level is a read-out (reporter) of all processes that has led to the ROS level being what it is in the cell.

Secondly, our claim - a mathematical claim - is that knowing only the cell's ROS level is sufficient for determining that cell's behavior: the cell's likelihood of duplicating, dying, and increasing in size. Saying this is different from saying that ROS is the only factor that determines a cell's behavior and that no other process in a cell affects cell's growth. Saying that ROS is the primary determinant of certain behaviors of a cell does NOT logically imply that ROS is the only mechanism that affects those behaviors. It just means that ROS level - which may report on many intracellular processes and many processes can indeed affect it - is by itself enough to let us determine the cell's behavior.

To be clear, we're saying that there is a mathematical function "F" which is: $F([\text{ROS}]) = \text{probability of cell duplicating}$. We obtain this by measurements in Fig. 3. This function requires only one input - the ROS concentration in a cell - to tell us that cell's behavior. If knowing ROS alone isn't sufficient for determining that cell's behavior, we wouldn't see the correlations that we observed between ROS level and the cell's behaviors in Fig. 3b. Our data indeed shows that intracellular ROS-concentration is the primary factor that determines a cell's survival, duplication ability, size growth, and death. **An integral part of our claim is that stochasticity is inevitable in a living system and that it leads to cell-to-cell variations – this stochasticity is an explicit part of our model (see the incorporation of noise in the derivation of our model in the Supplementary Notes).**

Given these reasons, **we agree with the reviewer's last sentence here as being true** (i.e., ROS **is** the primary determinant of growth inhibition).

If the reviewer's last sentence here is incorrect (i.e., reviewer's claim is that ROS is still not the primary factor for growth inhibition), we repeat our argument above: many things can alter the ROS level in a cell. Our manuscript does not state otherwise anywhere. Saying that knowing the ROS level alone is sufficient for determining the cell's behavior does not mean that there's only one mechanism that changes ROS level. We are **not** saying that there cannot be some other factors that inhibit growth inhibition at low temperatures. We **are** saying that even if there are some as-yet unknown factors that affect growth at low temperatures, ROS alone is

sufficient for us to use to predict growth behaviors of yeast at near-freezing temperatures **and** that these other factors are, at least partially, reported by the ROS level in as-yet unknown ways.

As just one of many examples in our work which highlights ROS as the primary factor: please look at the growth behaviors of mutants in Fig. 4e. The mutants that are by far the most affected are those that lack ROS-reducing enzymes (the two left-most green bars in Fig. 4e). We do not exclude the possibility of other effects possibly playing a role. Indeed, Fig. 4e also shows that knocking out Msn2/4 also causes growth defect at low temperatures. Intriguingly, some of the the ROS-reducing enzymes are controlled by the Msn2/4 stress response. We do not claim anywhere in our manuscript that no other factors besides ROS play a role in inhibiting growth. We're simply saying that knowledge of intracellular ROS levels alone is sufficient to determine growth behaviors of yeast at low temperatures (i.e., it is the (or one of the) main determinants). **In fact, this is the most conservative claim that we can make from our data.** The more radical statement would be to pick out one mechanism that changes ROS levels and then say that this is the only mechanism that inhibits growth. Our manuscript doesn't say this anywhere.

"4. Fig 2 Nutrients, p.7. I found this part quite speculative. It is quite obvious that adding antioxidants improves the growth rate, not only at 5 degrees. It was shown in many biological models, aging phases and conditions.

Please see our next response for why we disagree that this part is speculative. We believe that there's an unfortunate misunderstanding here (same misunderstanding of the logic as in the previous comment). We agree that adding antioxidants increases the growth rate as our data show. Importantly, we do not claim anywhere that we are the first ones to show that antioxidants can increase growth rate. In fact, even our earlier work (high-temperature paper) establishes a positive impact on growth by glutathione. So, we wouldn't be trying to win points for novelty by making such a claim here. We have not done so.

"Manipulation of the growth medium will definitely alter the metabolism, respiration, and oxidation rate. Reductive conditions increase doubling time in normal conditions as well."

Respectfully, we point out that the last sentence in the reviewer's comment is incorrect: "reductive" is "antioxidative", which would correspond to a decrease in the doubling time of a population, not an increase in the doubling time that the reviewer claims here. Our data clearly show that less ROS means a shorter doubling time on both the population-level and single-cell

level - because less cells die, more cells duplicate, and the cells that duplicate take less time to do so.

Aside from the above correction, we agree with the reviewer's statement that changing the growth medium alters many things including metabolism, oxidation rate, etc. Our experiment of changing the nutrient concentration in Fig. 2 or the contents of our manuscript do not contradict any of this. We do not state anywhere that none of these things change when we alter the growth medium.

We believe that there's an unfortunate, underlying misunderstanding here for the same reason as in the reviewer's comment #3 above: we are not saying that nothing besides ROS matters just because we can use [ROS] alone to predict cell's behaviors. In fact, even our own work shows that gene-expression speed (protein-synthesis rate) is required, independently of ROS, to determine the cell's doubling time. Similarly, we do not claim that removing ROS decreases the doubling time only at frigid temperatures and not at 30 °C. Decreasing the nutrient level at 30 °C (the reviewer's "normal condition") decreases the growth rate of yeast. This is a well-established phenomenon that is seen in many papers.

Many pathways and effects can funnel down to changing the level of ROS in a cell. We do not claim otherwise anywhere in our manuscript. Saying that [ROS] is the primary determinant (predictor) only means that regardless of the numerous mechanisms that affect ROS at low temperatures, knowing the [ROS] alone lets us predict many of the cell's growth and survival behaviors at the near-freezing temperatures.

In the case of the nutrient-changing experiment in Fig. 2, we are simply controlling [ROS] by decreasing the amount of nutrients – which indeed also affects metabolism and many things in yeast – and by changing the amount GSH that we add to the growth media. Both extracellular GSH and nutrient levels affect many other cellular processes, but those are not the main factors required to predict the population doubling time at frigid temperatures: knowing ROS is enough. Please note that nowhere in this argument, do we exclude - and there isn't any need to exclude - anything about metabolism changing in cells when nutrients changing, etc.

"Most probably, these conditions alter the diauxic shift of the yeast cells. Thus, I find the statement "these results show that non-sugar nutrients are major generators of ROS in frigid temp." is not accurate and also not solid."

Our results are not due to a diauxic shift because:

1. During a diauxic shift, yeast transitions to growing more slowly - there would be a slope-change in the growth curve as a function of time - because yeast switches to a different nutrient source (e.g., due to a more favorable sugar such as glucose being depleted). We do not see this in our data. On the contrary, in the nutrient-changing experiment shown in Figs. 2f-2i, we explicitly say that we **kept a saturating level of**

- glucose (2%) while collectively changing the level of non-sugar nutrients** (e.g., the pool of essential amino acids whose respective ratios were held fixed, but their total concentration was changed together, all while keeping [glucose] = 2%).
2. The population doubling times in Figs. 2g-i were measured during log-phase growth (i.e., in a population whose density was exponentially increasing over time by many orders of magnitude for many weeks before a diauxic shift occurs).
 3. **We provided direct, cell-free measurements** showing that changing the non-sugar nutrient level - while keeping a saturating level of glucose (2%) - changes the extracellular oxidation rate (Fig. 2f and Supplementary Fig. 13). This means that the non-sugar nutrients, either directly or indirectly through intracellular mechanisms, generate ROS.

Given these internally consistent results that we established through multiple techniques and in multiple conditions, we disagree that our statement, "these results show that non-sugar nutrients are major generators of ROS", is neither accurate nor solid. In fact, we respectfully point out that the reviewer states the following earlier in his/her major point #4: "*Manipulation of the growth medium will definitely alter the ... oxidation rate.*" **Hence, on the contrary, the reviewer contradicts themselves here, and our statement confirms the reviewer's earlier point.**

"In addition, the phrase "ROS scavengers (superoxide) " is very confusing. Superoxides are ROS. The method part is also not clear."

"ROS scavenger" means scavenger of ROS (e.g., scavenger of superoxides). Thank you for pointing out this confusion. **We changed "ROS scavenger" to "scavenger of ROS" in Fig. 2f and its associated text.**

"Minor comments:

1. The Introduction and abstract : there are many speculative statements and not clear phrases that do not reflect the paper.

What do "system level design principles that dictate the pace of life... " really mean.

We rewrote the first paragraph of the Introduction and the abstract so that it no longer mentions "design principles" and more clearly states the true intention of our work: using the budding yeast to address whether there are limits to thermally slowing down replicative dynamics of cells (eukaryotic cell-cycling) by decreasing temperature. This is an important and broad question that has fascinated physicists, quantitative biologists, and systems biologists. The revised introduction does a better job than the previous version of the Introduction in framing this as the true question and purpose for our work.

Our manuscript no longer mentions "design principles" anywhere. We previously used this term in a way that other systems biologists use it in myriad papers and books (e.g., Uri Alon's book titled "An introduction to systems biology: Design principles of biological circuit"). By design principles, we mean that cell-cycling time - the time taken by the cell to duplicate itself by fully completing the cell-cycle - is a function of many genes (e.g., genes involved in building a new cell) and many different chemical processes. So, our manuscript is asking a physicist's question of how to quantitatively derive the cell-cycling time from all these different chemical rates. That was our intention. But we no longer use this term in the manuscript.

"we discovered "key reason" - I am still not really convinced that ROS is the key reason. It can be a result of stress conditions, failure in metabolism and many reasons. ROS was measured but it is not the key reason for survival at 5 degrees."

"we discovered that all these effects of ROS are due to one mechanism: ROS elongates the G1 .." – again, there is more than one mechanism for sure, and maybe elongation of G1 results in increased ROS. The experiments presented here do not show direct regulation."

Respectfully, we again clarify this same, misunderstanding that is the basis for the previous comments.

We found a mathematical function "F" whereby: $F([ROS]) = \text{probability of a cell duplicating}$
We also found other mathematical functions like F that describe cell's other behaviors (size growth, death, doubling time, etc.).

This mathematical function "F" only takes in one input. The input is ROS level in a cell. With this one input, the function tells us the probability of that cell, with that much ROS, duplicating. This prediction is experimentally confirmed (Fig. 6). Hence ROS level in a cell - knowing only that - is sufficient to determine the cell's probability of duplicating.

This does NOT mean that there's only one mechanism that's important for cell duplicating or that there is only one way to control ROS. For example, our own work showed that we need the gene-expression speed, along with [ROS], to determine the cell's doubling time and speed limits. As another example, we showed that knocking out Whi5 lets cells with higher [ROS] enter the S-G2-M phase to duplicate. Moreover, our own work shows multiple ways of controlling [ROS]: these include changing the nutrient level (Fig. 2), adding GSH to growth medium (Fig. 2), knocking out ROS-reducing enzymes (Fig. 4), etc.

We agree with the reviewer and our manuscript demonstrates that many factors affect [ROS] in a cell. Multiple effects do funnel into affecting [ROS]. Many processes don't affect [ROS]. We don't state anything to the contrary anywhere in our manuscript. In the end, [ROS] in a cell results from all the processes that affect it. Mathematically, it alone is sufficient to predict the cell's behavior: knowing [ROS] in a cell alone is enough for us to predict the cell's behavior in a population at frigid temperatures. Experimentally changing [ROS] leads to outcomes that are consistent with all this. We explicitly reiterate this point in the summary of this work (last paragraph of introduction) and in the discussion so that the readers don't read further than what our actual claim is. **We also mention this in the Supplementary Discussion section (Pg. 85 of Supplementary Information) just to be crystal clear.**

"2. "Fig S12: "oxydation" should be oxidation."

Thank you for spotting this typo. **We corrected it (now Supplementary Fig. 13).**

"3. "laboratory-standard ("wt") yeast strain – there are many "standard" strains. Should be specified."

We now specify the strain (W303) in both the main text and in the Methods.

"4. Fig 3 what is the difference between "begins life" and "duplicates" – look very similar, both are in the budding phase."

"Begins life" means that we have a newborn cell whose bud neck snaps off – the daughter cell "begins life" (Fig. 3a - right snapshot in yellow box). "Duplicates" means that we see a mother cell beginning to duplicate by forming a bud (Fig. 3a - left snapshot in purple box) and ending duplication when that bud becomes the daughter (Fig. 3a - right snapshot in purple box) – the mother "duplicates". These are two different events that both involve a bud, one is about the newborn daughter cell ("begins life") and the other is about the mother that duplicates ("duplicates"). For "begins life" in Fig. 3a, we focus on the new daughter cell, the important moment is when the bud snaps (when life of the newborn daughter begins). We measure the ROS concentration in that just-born daughter cell and do so for many just-born daughter cells to obtain the probability for a cell to begin its life with a particular [ROS] (yellow curve in Fig. 3c). For the "duplicates" event, we're interested in the mother cell that begins to duplicate. We measure

the intracellular ROS concentration in the mother cell just when it starts to duplicate (when it forms a bud). We do so for many mother cells to obtain the probability of a cell to start duplication when it has a particular [ROS] (purple curve in Fig. 3c).

Thank you for taking the time to evaluate our manuscript and for finding our work to be "*interesting*", "*comprehensive*", "*very well written, easy to follow*", and to contain "*extensive supplementary figures to explain the data*".

We believe that many of the reviewer's comments arose from a misunderstanding of our mathematical and logical reasoning. In the past 2 months, aside from expanding our manuscript to include the additional data, we revised the manuscript (within the word-count limits) to avoid such misunderstandings, especially by adding the "Supplementary Discussion" section that paraphrases our responses. We hope that the revised manuscript and our explanations in this letter have resolved these misunderstandings.

REVIEWERS' COMMENTS

Reviewer #1 (Remarks to the Author):

Many thanks to the authors for addressing my previous suggestions. I still enthusiastically recommend publication of this manuscript. As a minor suggestion, it may make sense to note in the legend for Supplemental Figure S24a that they're no longer able to track the cell. It looks like this may also occur in Figure 4a. However, this is a very minor suggestion and should not delay publication of this exciting work.

Reviewer #2 (Remarks to the Author):

As mentioned in my previous comments, I thought the paper was ready to be published as it was. The paper is further strengthened in clarity and rigor with the additional experiments and revisions. It is the most comprehensive study I've reviewed (for different journals) in the last several years.

Response to Reviewer 1

Below, *the reviewer's comments are in italics* and are followed by our response.

" Many thanks to the authors for addressing my previous suggestions. I still enthusiastically recommend publication of this manuscript. As a minor suggestion, it may make sense to note in the legend for Supplemental Figure S24a that they're no longer able to track the cell. It looks like this may also occur in Figure 4a. However, this is a very minor suggestion and should not delay publication of this exciting work."

We are very grateful for this positive, overall assessment of our work. Following your suggestion, we now mention this in the caption for Supplementary Fig. 24a. But in the caption for Fig. 4a, we don't mention this because the situation is different (i.e., in fact, we continue to observe this cell but we cut the movie strip short for the sake of presentation).

Response to Reviewer 2

Below, *the reviewer's comments are in italics* and are followed by our response.

" As mentioned in my previous comments, I thought the paper was ready to be published as it was. The paper is further strengthened in clarity and rigor with the additional experiments and revisions. It is the most comprehensive study I've reviewed (for different journals) in the last several years."

Thank you so much for taking the time to evaluate our manuscript and for your very kind words.